# Neural Processes with Stability

**Huafeng Liu, Liping Jing, Jian Yu**
Beijing Key Lab of Traffic Data Analysis and Mining, Beijing Jiaotong University
The School of Computer and Information Technology, Beijing Jiaotong University
{hfliu1, lpjing, jianyu}@bjtu.edu.cn

## Abstract

Unlike traditional statistical models depending on hand-specified priors, neural processes (NPs) have recently emerged as a class of powerful neural statistical models that combine the strengths of neural networks and stochastic processes. NPs can define a flexible class of stochastic processes well suited for highly non-trivial functions by encoding contextual knowledge into the function space. However, noisy context points introduce challenges to the algorithmic stability that small changes in training data may significantly change the models and yield lower generalization performance. In this paper, we provide theoretical guidelines for deriving stable solutions with high generalization by introducing the notion of algorithmic stability into NPs, which can be flexible to work with various NPs and achieves less biased approximation with theoretical guarantees. To illustrate the superiority of the proposed model, we perform experiments on both synthetic and real-world data, and the results demonstrate that our approach not only helps to achieve more accurate performance but also improves model robustness.

## 1 Introduction

Neural processes (NPs) [9, 10] constitute a family of variational approximation models for stochastic processes with promising properties in computational efficiency and uncertainty quantification. Different from traditional statistical modeling for which a user typically hand-specifies a prior (e.g., smoothness of functions quantified by a Gaussian distribution in Gaussian process [25]), NPs implicitly define a broad class of stochastic processes with neural networks in a data-driven manner. When appropriately trained, NPs can define a flexible class of stochastic processes well suited for highly non-trivial functions that are not easily represented by existing stochastic processes.

NPs meta-learn a distribution over predictors and provide a way to select an inductive bias from data to adapt quickly to a new task. Incorporating the data prior into the model as an inductive bias, NPs can reduce the model complexity and improve model generalization. Usually, an NP predictor is described as predicting a set of data (target set) given a set of labeled data (context set). However, the number of noise in data introduces challenges to the algorithmic stability. In NPs, models are biased to the meta-datasets (a dataset of datasets), so small changes in the dataset (noisy or missing) may significantly change the models. As demonstrated in previous work [9, 10, 14, 11], existing NPs cannot provide stable predictions under noisy conditions, which may introduce high training error variance, and minimizing the training error may not guarantee consistent error reduction on the test set, i.e. low generalization performance [2]. In this case, algorithmic stability and generalization performance have strong connections and an unstable NP model has low generalization performance.

A stable model is one for which the learned solution does not change much with small changes in training set [2]. In general, heuristic techniques, such as cross-validation and ensemble learning, can be adopted to improve the generalization performance. Cross-validation needs to sacrifice the limited training data, while ensemble learning is computationally expensive on training sub-models. Recently, there are several improved NPs focused on considering model stability and

improving generalization performance empirically, such as hierarchical prior [27], stochastic attention mechanism [15], bootstrap [13], and Mixture of Expert [28]. However, most of them are unable to investigate the theoretical bound of the generalization performance of NPs. It is desirable to develop robust algorithms with low generalization error and high efficiency.

In this paper, we investigate NP-related models and explore more expressive stability toward general stochastic processes by proposing a stable solution. Specifically, by introducing the notion of stability into NPs, we focus on developing theoretical guidelines for deriving a stable NPs solution. We propose a method to find out subsets that are harder to predict than average, which is a key step for constructing this optimization problem. Based on it, a new extension of NPs with stable guarantees is formulated, which can be flexible to work with various NPs and achieves less biased approximation with theoretical guarantees. Considering the model adaptivity, an adaptive weighting strategy is proposed. To illustrate the superiority of the proposed stable solution, we perform experiments on synthetic 1D regression, system identification of physics engines, and real-world image completion tasks, and the results demonstrate that NPs with our stable solution are much more robust than original NPs.

## 2 Related Work

In this section, we briefly review two different areas which are highly relevant to the proposed method, neural processes, and algorithmic stability.

**Neural Processes** Neural processes are a well-known member of the stochastic process family by directly capturing uncertainties with deep neural networks, which are not only computationally efficient but also retain a probabilistic interpretation of the model [9, 10, 14, 13]. Starting with conditional neural processes (CNP) [9], there have been several follow-up works to improve NPs in various aspects [6]. Vanilla CNP combines neural networks with the Gaussian process to extract prior knowledge from training data. NP [10] introduces a global latent variable to model uncertainty in a variational manner. Considering the problem of underfitting in the vanilla NP, Attentive NP [14] introduces the attention mechanism to improve the model's reconstruction quality. [11] introduced convolutional conditional neural process (CONVCNP) models translation equivariance in the data. Wang and Van Hoof [27] presented a doubly stochastic variational process (DSVNP), which combines both global and local latent variables. Lee et al. [18] extended NP using Bootstrap and proposed the bootstrapping neural processes (BNP). Kawano et al. [13] presented a group equivariant conditional neural process by incorporating group equivariant into CNPs in a meta-learning manner. Wang and van Hoof [28] proposed to combine the Mixture of Expert models with NPs to develop more expressive exchangeable stochastic processes. Kim et al. [15] proposed a stochastic attention mechanism for NPs to capture appropriate context information. Although there are many NP variants to improve the model performance, those do not consider stability to yield high generalization performance.

**Algorithmic Stability** Stability, as known as algorithmic stability, is a computational learning theory of how a machine learning algorithm is perturbed by small changes to its inputs [2]. Many efforts have been made to analyze various notions of algorithmic stability and prove that a broad spectrum of learning algorithms are stable in some sense [2, 3, 29, 12]. [3] proved that $l_2$ regularized learning algorithms are uniformly stable and able to obtain new bounds on generalization performance. [29] generalized [3]'s results and proved that regularized learning algorithms with strongly convex penalty functions on bounded domains. Hardt et al. [12] showed that parametric models trained by stochastic gradient descent algorithms are uniformly stable. Li et al. [19] introduced the stability notation to low-rank matrix approximation. Liu et al. [22] proved that tasks in multi-task learning can act as regularizers and that multi-task learning in a very general setting will therefore be uniformly stable under mild assumptions. This is the first work to investigate the stability of NPs from theoretical guidelines and derive NPs solutions with high stability.

## 3 Preliminary

Let calligraphic letters (e.g., $\mathcal{A}$) indicate sets, capital letters (e.g., $A$) indicate scalars, lower-case bold letters (e.g., $\mathbf{a}$) denote vectors, and capital bold letters (e.g., $\mathbf{A}$) indicate matrices. Suppose there is a dataset $\mathcal{D} = (\mathbf{X}, \mathbf{y}) = \{(\mathbf{x}_i, y_i)\}_{i=1}^N$ with $N$ data points $\mathbf{X} = [\mathbf{x}_1, \mathbf{x}_2, \cdots, \mathbf{x}_N]^\top \in \mathbb{R}^{N \times D}$, and corresponding labels $\mathbf{y} = [y_1, y_2, \cdots, y_N] \in \mathbb{R}^N$. Considering an arbitrary number of data points $\mathcal{D}^{\mathcal{C}} = (\mathbf{X}^{\mathcal{C}}, \mathbf{y}^{\mathcal{C}}) = \{(\mathbf{x}_i, y_i)\}_{i \in \mathcal{C}}$, where $\mathcal{C} \subseteq \{1, 2, \cdots, N\}$ is an index set defining context information, neural processes model the conditional predictive distribution of the target values $\mathbf{y}^{\mathcal{T}} = \{y_i\}_{i \in \mathcal{T}}$

at some target data points $\mathbf{X}^{\mathcal{T}} = \{\mathbf{x}_i\}_{i \in \mathcal{T}}$ based on the context $\mathcal{D}^{\mathcal{C}}$, i.e. $P(\mathbf{y}^{\mathcal{T}}|\mathbf{X}^{\mathcal{T}}, \mathcal{D}^{\mathcal{C}})$. Usually, target set is defined as $\mathcal{T} = \{1, 2, \cdots, N\}$. Only in CNP [9], $\mathcal{T} \subseteq \{1, 2, \cdots, N\}$ and $\mathcal{T} \cap \mathcal{C} = \emptyset$. In this paper, we define $\mathcal{T} = \{1, 2, \cdots, N\}$ for all NPs, i.e. conditional predictive distribution is $P(\mathbf{y}|\mathbf{X}, \mathcal{D}^{\mathcal{C}}) = \prod_{i=1}^{N} P(y_i|\mathbf{x}_i, \mathcal{D}^{\mathcal{C}})$.

Fundamentally, there are two NP variants: deterministic and probabilistic. Deterministic NP [9], i.e. CNP, models the conditional distribution as $P(\mathbf{y}|\mathbf{X}, \mathcal{D}^{\mathcal{C}}) = P(\mathbf{y}|\mathbf{X}, \mathbf{r}^{\mathcal{C}})$, where $\mathbf{r}^{\mathcal{C}} \in \mathbb{R}^d$ is an aggregated feature vector processed by a function that maps $\mathcal{D}^{\mathcal{C}}$ into a finite-dimensional vector space in a permutation-invariant way. In probabilistic NPs [10], a latent variable $\mathbf{z} \in \mathbb{R}^d$ is introduced to capture model uncertainty and the NPs infer $P_\theta(\mathbf{z}|\mathcal{D}^{\mathcal{C}})$ given context set using the reparameterization trick [16] and models such a conditional distribution as $P_\theta(y_i|\mathbf{x}_i, \mathcal{D}^{\mathcal{C}}) = \int P_\theta(y_i|\mathbf{x}_i, \mathcal{D}^{\mathcal{C}}, \mathbf{z}) P_\theta(\mathbf{z}|\mathcal{D}^{\mathcal{C}}) d\mathbf{z}$ and it is trained by maximizing an ELBO: $\mathbb{E}_{\mathbf{z} \sim P_\theta(\mathbf{z}|\mathbf{X}, \mathbf{y})}[\log P_\theta(\mathbf{y}|\mathbf{X})] - KL[P_\theta(\mathbf{z}|\mathbf{X}, \mathbf{y}) \| P_\theta(\mathbf{z}|\mathcal{D}^{\mathcal{C}})]$.

**Meta Training NP Prediction** To achieve fast prediction on a new context set at test time, NPs meta-learn a distribution over predictors. To perform meta-learning, we require a meta-dataset (dataset of datasets). We consider an unknown distribution $\mu$ on an instance space $\mathcal{X} \times \mathcal{Y}$, and a set of independent sample $\mathcal{D} = \{(\mathbf{x}_i, y_i)\}_{i=1}^{N}$ drawn from $\mu$: $(\mathbf{x}_i, y_i) \sim \mu$ and $\mathcal{D} \sim \mu^N$. Suppose meta-dataset contains $M$ datasets $\mathcal{D}_{1:M} = \{\mathcal{D}_m\}_{m=1}^{M}$ with $\mathcal{D}_m = \{\mathcal{D}_m^{\mathcal{C}}, \mathcal{D}_m^{\mathcal{T}}\}$, we assume that all $M$ datasets drawn from a common environment $\tau$, which is a probability measure on the set of probability measures on $\mathcal{X} \times \mathcal{Y}$. The draw of $\mu \sim \tau$ indicates the encounter of a specific learning task $\mu$ in the environment $\tau$. For simplicity, we assume that each dataset has the same sample size $N$. Following the previous work related to multi-task learning [24] and meta learning [5], The environment $\tau$ induces a measure $\mu_{N,\tau}$ on $(\mathcal{X} \times \mathcal{Y})^N$ such that $\mu_{N,\tau}(A) = \mathbb{E}_{\mu \sim \tau}[\mu^N(A)], \forall A \subseteq (\mathcal{X} \times \mathcal{Y})^N$. Thus a dataset $\mathcal{D}_m$ is independently sampled from a task $\mu$ encountered in $\tau$, which is denoted as $\mathcal{D}_m \sim \mu_{N,\tau}$ for $m \in [M]$.

Suppose there exists a meta parameter $\theta$ indicating the shared knowledge among different tasks. In this case, a meta learning algorithm $\mathcal{A}_{meta}$ for NPs takes meta-datasets $\mathcal{D}_{1:M}$ as input, and then outputs a meta parameter $\theta = \mathcal{A}_{meta}(\mathcal{D}_{1:M}) \sim P_{\theta|\mathcal{D}_{1:M}}$. When given a new test dataset $\mathcal{D}$, we can evaluate the quality of the meta parameter $\theta$ by the following true risk:

$$R_\tau(\theta) = \mathbb{E}_{\mathcal{D} \sim \mu_{N,\tau}} \mathbb{E}_{U \sim P_{\theta|\mathcal{D}_{1:M}}} [R_\mu(\theta)] \tag{1}$$

where $R_\mu(\theta) = -\mathbb{E}_{(\mathbf{x}_i, y_i) \sim \mu} \log P_\theta(y_i|\mathbf{x}_i, \mathcal{D}^{\mathcal{C}})$. Usually, $\tau$ and $\mu$ are unknown, we can only estimate the meta parameter $\theta$ from the observed data $\mathcal{D}_{1:M}$. In this case, the empirical risk w.r.t $\theta$ is:

$$R_{\mathcal{D}_{1:M}}(\theta) = 1/M \sum_{m=1}^{M} \mathbb{E}_{\theta \sim P_{\theta|\mathcal{D}_m^{\mathcal{C}}}} R_{\mathcal{D}_m}(\theta) \tag{2}$$

where $R_{\mathcal{D}_m}(\theta) = -(1/N) \sum_{i=1}^{N} \log P_\theta(y_i|\mathbf{x}_i, \mathcal{D}^{\mathcal{C}})$.

NPs have various strengths: 1) *Efficiency*: meta-learning allows NPs to incorporate information from a new context set and make predictions with a single forward pass. The complexity is linear or quadratic in the context size instead of cubic as with Gaussian process regression; 2) *Flexibility*: NPs can define a conditional distribution of an arbitrary number of target points, conditioning an arbitrary number of observations; 3) *Permutation invariance*: the encoders of NPs use set property [32] to make the target prediction permutation invariant. Thanks to these properties, NPs are widely-used in lots of tasks, e.g., Bayesian optimization [8], recommendation [20, 21], physics engines controlling [27] etc. While there are many NP variants to improve the performance of NPs [9, 10, 14, 13, 15, 28], those do not take model's stability into consider account yet, which is the key to the robustness of the model.

## 4 Problem Formulation

**Stability of NP** A stable learning algorithm has the property that replacing one element in the training set does not result in a significant change to the algorithm's output [2]. Therefore, if we take the training error as a random variable, the training error of a stable learning algorithm should have a small variance. This implies that stable algorithms have the property that the training errors are close to the testing error [2]. Based on the defined risks, the algorithmic stability of approximate $\{y_i\}_{i \in \mathcal{T}}$ in NPs is defined as follows.

**Definition 4.1.** (Algorithmic Stability of Neural Processes) For any measure $\mu_{N,\tau}$ on $(\mathcal{X} \times \mathcal{Y})^N$ such that $\mu_{N,\tau}(A) = \mathbb{E}_{\mu \sim \tau}[\mu^N(A)], \forall A \subseteq (\mathcal{X} \times \mathcal{Y})^N$, sample $M$ datasets $\mathcal{D}_{1:M}$ from $\mu_{N,\tau}$ randomly. For a given $\epsilon > 0$, we say that $R_{\mathcal{D}_{1:M}}(\theta)$ is $\delta$-stable if the following holds:

$$P\left(|R_\tau(\theta) - R_{\mathcal{D}_{1:M}}(\theta)| \leq \epsilon\right) \geq 1 - \delta. \tag{3}$$

The above stability for NPs has the property that the generalization error is bounded, which indicates that minimizing the training error will have a high probability of minimizing the testing error. This new stability notion makes it possible to measure the generalization performance between different NP approximations. For instance, for any two meta-datasets $\mathcal{D}^1_{1:M}$ and $\mathcal{D}^2_{1:M}$ from $\mu_{N,\tau}$, train NPs on $\mathcal{D}^1_{1:M}$ and $\mathcal{D}^2_{1:M}$ are $\delta_1$-stable and $\delta_2$-stable, respectively. Then $R_{\mathcal{D}^1_{1:M}}(\theta)$ is more stable than $R_{\mathcal{D}^2_{1:M}}(\theta)$ if $\delta_1 < \delta_2$. This implies that $R_{\mathcal{D}^1_{1:M}}(\theta)$ is close to $R_\tau(\theta)$ with higher probability than $R_{\mathcal{D}^2_{1:M}}(\theta)$, i.e. minimizing $R_{\mathcal{D}^1_{1:M}}(\theta)$ will lead to solutions that are of high probabilities with better generalization performance than minimizing $R_{\mathcal{D}^2_{1:M}}(\theta)$.

Based on the above analysis, we can see that the reliability of data points is crucial to the success of NPs and frail NPs are susceptible to noise.

**Stability vs. Generalization Error** The sparsity of the data, incomplete and noisy introduces challenges to the algorithm stability. NP models are biased to the quality of context data and target data, so small changes in the training data (noisy) may significantly change the models. In this case, unstable solutions will introduce high training error variance, and minimizing the training error may not guarantee consistent error reduction on the testing dataset, i.e., low generalization performance. In other words, the algorithm stability has a direct impact on generalization performance, and an unstable NP solution has low generalization performance. We take NPs with 1D regression task as an example [9] to inves-

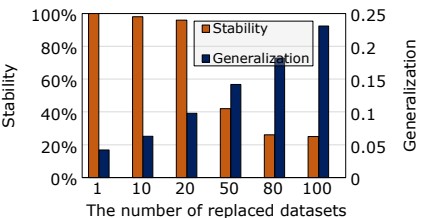

Figure 1: Stability vs. generalization error with different numbers of replaced noisy datasets.

tigate the relationship between generalization performance and stability of NPs. The total number of training and testing datasets is 200 and 100. We trained the NPs model with curves generated from the Gaussian process with RBF kernels by replacing the normal data dataset with a noisy dataset, i.e. the number of replaced datasets is turned in $\{1, 10, 20, 50, 80, 100\}$. We quantify stability changes of NPs with the generalization error when the number of replaced datasets increases from 1 to 100. We compute the difference between training error and test error to measure generalization error. We define the difference between test error and training error as $R_\tau(\theta) - R_{\mathcal{D}_{1:M}}(\theta)$, and compute $P(|R_\tau(\theta) - R_{\mathcal{D}_{1:M}}(\theta)| \leq \epsilon)$ with 100 different runs to measure stability. Here we choose $\epsilon$ in Definition 4.1 as 0.0015 to cover all error differences when the number of replaced datasets is 1. As shown in Figure 1, the generalization error increases when the number of replaced points increases since the testing error becomes lower. On the contrary, the stability of NP decreases with the number of replaced points increases. This indicates that stability decreases with generalization error increases. This study demonstrates that existing NPs suffer from lower generalization performance due to low algorithmic stability. Therefore, it is important to develop a stable solution for NPs that offers good generalization performance.

# 5 Method

In this section, inspired by the previous work [19], we present a stable solution for NPs with stability and high generalization. Algorithmic stability provides an intuitive way to measure the changes in the outputs of a learning algorithm when the input is changed. Various ways have been introduced to measure algorithmic stability. Following the definition of uniform stability [2], given a stable NP, the approximation results remain stable if the change of the datasets. For instance, we can remove a subset of easily predictable data points from $\mathcal{D}_{1:M}$ to obtain $\mathcal{D}'_{1:M}$. It is desirable that the solution of minimizing both $\mathcal{D}_{1:M}$ and $\mathcal{D}'_{1:M}$ together will be more stable than the solution of minimizing $\mathcal{D}_{1:M}$ only. The following Theorem formally proves the statement.

**Theorem 5.1.** *Let $\mathcal{D}_{1:M}$ ($M \geq 2$) be a sampled meta-dataset of measure $\mu_{N,\tau}$. Let $\mathcal{D}_s \in \mathcal{D}_{1:M}$ be a subset of the meta-dataset, which satisfy that $\forall(\mathbf{x}_i, y_i) \in \mathcal{D}_s$, $-\log P_\theta(y_i|\mathbf{x}_i, \mathcal{D}^C) \leq R_{\mathcal{D}_{1:M}}(\theta)$. Let $\mathcal{D}'_{1:M} = \mathcal{D}_{1:M} - \mathcal{D}_s$, then for any $\epsilon > 0$ and $1 > w_0 > 0$, $1 > w_1 > 0$ ($w_0 + w_1 = 1$), $w_0 R_{\mathcal{D}_{1:M}}(\theta) + w_1 R_{\mathcal{D}'_{1:M}}(\theta)$ and $R_{\mathcal{D}_{1:M}}(\theta)$ are $\delta_1$-stable and $\delta_2$-stable, respectively, then $\delta_1 \leq \delta_2$.*

*Proof.* Let's assume that $R_\tau(\theta) - R_{\mathcal{D}_{1:M}}(\theta) \in [-a_1, a_1]$ and $R_\tau(\theta) - (w_0 R_{\mathcal{D}_{1:M}}(\theta) + w_1 R_{\mathcal{D}'_{1:M}}(\theta)) \in [-a_2, a_2]$ are two random variables with zero mean, where $a_1 = \sup\{R_\tau(\theta) - R_{\mathcal{D}_{1:M}}(\theta)\}$ and $a_2 = $

$\sup\{R_\tau(\theta) - (w_0 R_{\mathcal{D}_{1:M}}(\theta) + w_1 R_{\mathcal{D}'_{1:M}}(\theta))\}$. Based on Markov's inequality[1], for any $t > 0$, we have

$$P(R_\tau(\theta) - R_{\mathcal{D}_{1:M}}(\theta) \geq \epsilon) \leq \frac{\mathbb{E}\left[e^{t\left(R_\tau(\theta) - R_{\mathcal{D}_{1:M}}(\theta)\right)}\right]}{e^{t\epsilon}}. \tag{4}$$

Based on Hoeffding's lemma[2], we have $\mathbb{E}[e^{t(R_\tau(\theta) - R_{\mathcal{D}_{1:M}}(\theta))}] \leq e^{\frac{1}{2}t^2 a_1^2}$, i.e. $P(R_\tau(\theta) - R_{\mathcal{D}_{1:M}}(\theta) \geq \epsilon) \leq \frac{e^{\frac{1}{2}t^2 a_1^2}}{e^{t\epsilon}}$. Similarly, we have $P(R_\tau(\theta) - R_{\mathcal{D}_{1:M}}(\theta) \leq -\epsilon) \leq \frac{e^{\frac{1}{2}t^2 a_1^2}}{e^{t\epsilon}}$. Combining those two inequalities, we have $P(|R_\tau(\theta) - R_{\mathcal{D}_{1:M}}(\theta)| \geq \epsilon) \leq \frac{2e^{\frac{1}{2}t^2 a_1^2}}{e^{t\epsilon}}$, i.e.

$$P(|R_\tau(\theta) - R_{\mathcal{D}_{1:M}}(\theta)| \leq \epsilon) \geq 1 - \frac{2e^{\frac{1}{2}t^2 a_1^2}}{e^{t\epsilon}}. \tag{5}$$

Similarly, we have

$$P\left(|R_\tau(\theta) - (w_0 R_{\mathcal{D}_{1:M}}(\theta) + w_1 R_{\mathcal{D}'_{1:M}}(\theta))| \leq \epsilon\right) \geq 1 - \frac{2e^{\frac{1}{2}t^2 a_2^2}}{e^{t\epsilon}}. \tag{6}$$

In this case, the relationship between $a_1$ and $a_2$ is

$$\begin{aligned}
a_2 &= \sup\left\{R_\tau(\theta) - R_{\mathcal{D}_{1:M}}(\theta) + w_1\left(R_{\mathcal{D}_{1:M}}(\theta) - R_{\mathcal{D}'_{1:M}}(\theta)\right)\right\} \\
&= \sup\left\{R_\tau(\theta) - R_{\mathcal{D}_{1:M}}(\theta)\right\} + w_1 \sup\left\{R_{\mathcal{D}_{1:M}}(\theta) - R_{\mathcal{D}'_{1:M}}(\theta)\right\} \\
&= a_1 + \lambda_1 \sup\left\{R_{\mathcal{D}_{1:M}}(\theta) - R_{\mathcal{D}'_{1:M}}(\theta)\right\}
\end{aligned} \tag{7}$$

Since $\forall(\mathbf{x}_i, y_i) \in \mathcal{D}_s$, $-\log P_\theta(y_i|\mathbf{x}_i, \mathcal{D}_s^{\mathcal{C}}) \leq R_{\mathcal{D}_{1:M}}(\theta)$, we have $-(1/N)\sum_{i=1}^N \log P_\theta(y_i|\mathbf{x}_i, \mathcal{D}_s^{\mathcal{C}}) \leq R_{\mathcal{D}_{1:M}}(\theta)$. Then, since $\mathcal{D}_{1:M} = \mathcal{D}_s \cup \mathcal{D}'_{1:M}$. This means that $\sup\{R_{\mathcal{D}_{1:M}}(\theta) - R_{\mathcal{D}'_{1:M}}(\theta)\} \leq 0$. Thus, we have $a_2 \leq a_1$. This turns out that $\frac{2e^{\frac{1}{2}t^2 a_2^2}}{e^{t\epsilon}} \leq \frac{2e^{\frac{1}{2}t^2 a_1^2}}{e^{t\epsilon}}$, i.e. $\delta_1 \leq \delta_2$. $\qquad\square$

Theorem 5.1 indicates that, if we remove a subset that is easier to predict than average from $\mathcal{D}_{1:M}$ to form $\mathcal{D}'_{1:M}$, then $w_0 R_{\mathcal{D}_{1:M}}(\theta) + w_1 R_{\mathcal{D}'_{1:M}}(\theta)$ has higher probability of being close to $R_\tau(\theta)$ than $R_{\mathcal{D}_{1:M}}(\theta)$. Therefore, minimizing $w_0 R_{\mathcal{D}_{1:M}}(\theta) + w_1 R_{\mathcal{D}'_{1:M}}(\theta)$ will lead to solutions that have better generalization performance than minimizing $R_{\mathcal{D}_{1:M}}(\theta)$.

However, Theorem 5.1 only proves that it is beneficial to remove an easily predictable dataset from $\mathcal{D}_{1:M}$ to obtain $\mathcal{D}'_{1:M}$, but does not show how many datasets we should remove from $\mathcal{D}_{1:M}$. Actually, removing more datasets that satisfy $-\log P_\theta(y_i|\mathbf{x}_i, \mathcal{D}^{\mathcal{C}}) \leq R_{\mathcal{D}_{1:M}}(\theta)$ can obtain better $\mathcal{D}'_{1:M}$, as shown in following Theorem 5.2.

**Theorem 5.2.** *Let $\mathcal{D}_{1:M}$ ($M \geq 2$) be a sampled meta-dataset of measure $\mu_{N,\tau}$. Let $\mathcal{D}_{s1}$ and $\mathcal{D}_{s2}$ be two subsets of $\mathcal{D}_{1:M}$, which satisfy $\mathcal{D}_{s2} \subset \mathcal{D}_{s1} \subset \mathcal{D}_{1:M}$. $\mathcal{D}_{s2}$ and $\mathcal{D}_{s1}$ satisfy that $\forall(\mathbf{x}_i, y_i) \in \mathcal{D}_{s1}$, $-\log P_\theta(y_i|\mathbf{x}_i, \mathcal{D}^{\mathcal{C}}) \leq R_{\mathcal{D}_{1:M}}(\theta)$. Let $\mathcal{D}^1_{1:M} = \mathcal{D}_{1:M} - \mathcal{D}_{s1}$ and $\mathcal{D}^2_{1:M} = \mathcal{D}_{1:M} - \mathcal{D}_{s2}$, then for any $\epsilon > 0$ and $1 > w_0 > 0$, $1 > w_1 > 0$ ($w_0 + w_1 = 1$), $w_0 R_{\mathcal{D}_{1:M}}(\theta) + w_1 R_{\mathcal{D}^1_{1:M}}(\theta)$ and $w_0 R_{\mathcal{D}_{1:M}}(\theta) + w_1 R_{\mathcal{D}^2_{1:M}}(\theta)$ are $\delta_1$-stable and $\delta_2$-stable, respectively, then $\delta_1 \leq \delta_2$.*

*Proof.* Let's assume that $R_\tau(\theta) - (w_0 R_{\mathcal{D}_{1:M}}(\theta) + w_1 R_{\mathcal{D}^1_{1:M}}(\theta)) \in [-a_1, a_1]$ and $R_\tau(\theta) - (w_0 R_{\mathcal{D}_{1:M}}(\theta) + w_1 R_{\mathcal{D}^2_{1:M}}(\theta)) \in [-a_2, a_2]$ are two random variables with $0$ mean, where $a_1 = \sup\{R_\tau(\theta) - (w_0 R_{\mathcal{D}_{1:M}}(\theta) + w_1 R_{\mathcal{D}^1_{1:M}}(\theta))\}$ and $a_2 = \sup\{R_\tau(\theta) - (w_0 R_{\mathcal{D}_{1:M}}(\theta) + w_1 R_{\mathcal{D}^2_{1:M}}(\theta))\}$.

Then, based on Markov's inequality and Hoeffding's lemma, we have

$$\begin{aligned}
P\left(\left|R_\tau(\theta) - \left(w_0 R_{\mathcal{D}_{1:M}}(\theta) + w_1 R_{\mathcal{D}^1_{1:M}}(\theta)\right)\right| \leq \epsilon\right) &\geq 1 - \frac{2e^{\frac{1}{2}t^2 a_1^2}}{e^{t\epsilon}}, \\
P\left(\left|R_\tau(\theta) - \left(w_0 R_{\mathcal{D}_{1:M}}(\theta) + w_1 R_{\mathcal{D}^2_{1:M}}(\theta)\right)\right| \leq \epsilon\right) &\geq 1 - \frac{2e^{\frac{1}{2}t^2 a_2^2}}{e^{t\epsilon}}.
\end{aligned} \tag{8}$$

Since $\forall(\mathbf{x}_i, y_i) \in \mathcal{D}_{s1}$, $-\log P_\theta(y_i|\mathbf{x}_i, \mathcal{D}^{\mathcal{C}}) \leq R_{\mathcal{D}_{1:M}}(\theta)$ and $\mathcal{D}_{s2} \subset \mathcal{D}_{s1} \subset \mathcal{D}_{1:M}$, we have $R_{\mathcal{D}^1_{1:M}}(\theta) \leq R_{\mathcal{D}^2_{1:M}}(\theta)$. Thus, we have $\sup\{R_{\mathcal{D}^1_{1:M}}(\theta) - R_{\mathcal{D}^2_{1:M}}(\theta)\} \leq 0$ since $a_1 = a_2 + w_1 \sup\{R_{\mathcal{D}^1_{1:M}}(\theta) - R_{\mathcal{D}^2_{1:M}}(\theta)\}$ and $a_1 \leq a_2$. Then, we can conclude that $\delta_1 \leq \delta_2$. $\qquad\square$

---

[1] (Extended version of Markov's Inequality) Let $x$ be a real-valued non-negative random variable and $\varphi(\cdot)$ be a nondecreasing nonnegative function with $\varphi(a) > 0$. Then, for any $\epsilon > 0$, $P(x \geq \epsilon) \leq \frac{\mathbb{E}[\varphi(x)]}{\varphi(\epsilon)}$.

[2] (Hoeffding's Lemma) Let $x$ be a real-valued random variable with zero mean and $p(x \in [a, b]) = 1$. Then, for any $z \in \mathbb{R}$, $\mathbb{E}[e^{zx}] \leq \exp\left(\frac{1}{8}z^2(b-a)^2\right)$.

Theorem 5.2 indicates that removing more data points that are easy to predict will obtain more stable NPs. Therefore, it is desirable to choose $\mathcal{D}'_{1:M}$ (i.e. $\mathcal{D}_{1:M} - \mathcal{D}_s$) as the whole set which is harder to predict than average, i.e. the whole set satisfying $\forall(\mathbf{x}_i, y_i) \in \mathcal{D}'_{1:M}, -\log P_\theta(y_i|\mathbf{x}_i, \mathcal{D}^\mathcal{C}) \leq R_{\mathcal{D}_{1:M}}(\theta)$. Without loss of generality, we can extend Theorem 5.2 by considering several harder predicted sets to obtain a more stable solution by minimizing them all together. In this case, we need to prove that the stability of a model with $K$ subsets is better than the model with $K-1$ subsets, as shown in Theorem 5.3.

**Theorem 5.3.** *Let $\mathcal{D}_{1:M}$ ($M \geq 2$) be a sampled meta-dataset of measure $\mu_{N,\tau}$. Let $\mathcal{D}_{s1}, \mathcal{D}_{s2}, \cdots, \mathcal{D}_{sK} \subset \mathcal{D}_{1:M}$ be $K$ subsets and satisfy $\forall(\mathbf{x}_i, y_i) \in \mathcal{D}_{sk}(k \in [K]), -\log P_\theta(y_i|\mathbf{x}_i, \mathcal{D}^\mathcal{C}) \leq R_{\mathcal{D}_{1:M}}(\theta)$. Let $\mathcal{D}^k_{1:M} = \mathcal{D}_{1:M} - \mathcal{D}_{sk}$ for all $k \in [K]$, then for any $\epsilon > 0$ and $1 > w_k > 0$ for all $k \in [K]$, $(w_0 + w_1 + \cdots + w_K = 1)$, $w_0 R_{\mathcal{D}_{1:M}}(\theta) + \sum_{k=1}^{K} w_k R_{\mathcal{D}^k_{1:M}}(\theta)$ and $(w_0 + w_K) R_{\mathcal{D}_{1:M}}(\theta) + \sum_{k=1}^{K-1} w_k R_{\mathcal{D}^k_{1:M}}(\theta)$ are $\delta_1$-stable and $\delta_2$-stable, respectively, then $\delta_1 \leq \delta_2$.*

*Proof.* For simplicity, we denote $w_0 R_{\mathcal{D}_{1:M}}(\theta) + \sum_{k=1}^{K} w_k R_{\mathcal{D}^k_{1:M}}(\theta)$ as $R_1$ and $(w_0 + w_K) R_{\mathcal{D}_{1:M}}(\theta) + \sum_{k=1}^{K-1} w_k R_{\mathcal{D}^k_{1:M}}(\theta)$ as $R_2$. Let's assume that $R_\tau(\theta) - R_1 \in [-a_1, a_1]$ and $R_\tau(\theta) - R_2 \in [-a_2, a_2]$ are two random variables with 0 mean, where $a_1 = \sup\{R_\tau(\theta) - R_1\}$ and $a_2 = \sup\{R_\tau(\theta) - R_2\}$. Then, based on Markov's inequality and Hoeffding's lemma, we have

$$P(|R_\tau(\theta) - R_1| \leq \epsilon) \geq 1 - \frac{2e^{\frac{1}{2}t^2 a_1^2}}{e^{t\epsilon}}, \qquad (|R_\tau(\theta) - R_2| \leq \epsilon) \geq 1 - \frac{2e^{\frac{1}{2}t^2 a_2^2}}{e^{t\epsilon}}. \tag{9}$$

Similar to the proof of Theorem 5.3, we have $R_{\mathcal{D}^K_{1:M}}(\theta) \geq R_{\mathcal{D}_{1:M}}(\theta)$, which indicates that $\sup\{R_{\mathcal{D}_{1:M}}(\theta) - R_{\mathcal{D}^K_{1:M}}(\theta)\} \leq 0$. Since $a_2 = a_1 + w_K \sup\{R_{\mathcal{D}_{1:M}}(\theta) - R_{\mathcal{D}^K_{1:M}}(\theta)\}$, we know that $a_2 \leq a_1$. Thus we can conclude that $\frac{2e^{\frac{1}{2}t^2 a_2^2}}{e^{t\epsilon}} \leq \frac{2e^{\frac{1}{2}t^2 a_1^2}}{e^{t\epsilon}}$, i.e. $\delta_1 \leq \delta_2$. $\qquad\square$

Based on Theorem 5.3, we know that optimization on $\mathcal{D}_{1:M}$ and more than one hard predictable subsets of $\mathcal{D}_{1:M}$ can achieve more stable prediction. However, how to select data points that are difficult to predict from $\mathcal{D}_{1:M}$ is still a challenging problem, especially, since we need to select $K$ subsets.

## 5.1 The Proposed Solution

According to the analysis of stability in NPs, we propose a stable solution for NPs to achieve model stability with the aid of hard predictable subsets selection. Specifically, we introduce a solution to obtain those hard predictable subsets based on *only one* set of easily predicted data points which can be broken into four steps:

(1) Selecting a existing NP model (e.g., CNP [9], NP[10]) and training it with meta-dataset $\mathcal{D}_{1:M}$;

(2) Selecting an easily predicted subset $\mathcal{D}_s \subset \mathcal{D}_{1:M}$, which satisfies that $\forall(\mathbf{x}_i, y_i) \in \mathcal{D}_s$, $-\log P_\theta(y_i|\mathbf{x}_i, \mathcal{D}^\mathcal{C}) \leq R_{\mathcal{D}_{1:M}}(\theta)$;

(3) Dividing $\mathcal{D}_s$ into $K$ non-overlapping subsets $\mathcal{D}_{s1}, \mathcal{D}_{s2}, \cdots, \mathcal{D}_{sK}$ and satisfying $\cup_{k=1}^{K} \mathcal{D}_{sk} = \mathcal{D}_s$;

(4) Defining $K$ subsets that are difficult to predict, i.e. $\mathcal{D}^k_{1:M} = \mathcal{D}_{1:M} - \mathcal{D}_{sk}$ for all $k \in [K]$;

Thus, a new extension of NPs is given as

$$\mathcal{L} = \arg\min_\theta w_0 R_{\mathcal{D}_{1:M}}(\theta) + \sum_{k=1}^{K} w_k R_{\mathcal{D}^k_{1:M}}(\theta), \tag{10}$$

where $w_0, w_1, \cdots, w_K$ indicate the contributions of each component and satisfy $\sum_{k=0}^{K} w_k = 1$.

The whole learning algorithm is given in Algorithm 1. From steps 1 to 12, we obtain $K$ different hard predictable subsets, and the complexity of lines 1 to 12 is $O(MN)$. The complexity of line 11 is related to the applied NP models (such as CNP, NP, ANP, etc). Thus, the computational complexity of NPs with our stable solution is similar to the original NPs. As shown in Algorithm 1, we need to pre-train the base model to select samples. In fact, the pre-trained model can not only be used for sample selection but its model parameters can be used as the initialization of the stable version model. At this time, the training of the stable version can converge faster.

**Stability Guarantee** Here we give a theoretical guarantee of the proposed stable solution.

**Theorem 5.4.** *Let $\mathcal{D}_{1:M}$ ($M \geq 2$) be a sampled meta-dataset of measure $\mu_{N,\tau}$. Let $\mathcal{D}_s \subset \mathcal{D}_{1:M}$ which satisfies that $\forall(\mathbf{x}_i, y_i) \in \mathcal{D}_s, -\log P_\theta(y_i|\mathbf{x}_i, \mathcal{D}^\mathcal{C}) \leq R_{\mathcal{D}_{1:M}}(\theta)$. By dividing $\mathcal{D}_s$ into $K$ subsets*

---

**Algorithm 1** Learning algorithm for stable NPs

---

**Input:** Meta-dataset $\mathcal{D}_{1:M}$, $\beta > 0.5$ is a predefined probability for data selection. $\mathcal{D}_s = \emptyset$.

1: Train a NP model with parameter $\theta$ based on $R_{\mathcal{D}_{1:M}}(\theta)$;
2: **for** $(\mathbf{x}_i, y_i) \in \mathcal{D}_{1:M}$ **do**
3:      randomly generate split parameter $\rho_i \in [0,1]$;
4:      **if** $((-\log P_\theta(y_i|\mathbf{x}_i, \mathcal{D}^{\mathcal{C}}) \leq R_{\mathcal{D}_{1:M}}(\theta))$ & $\rho_i \leq \beta)$
5:      or $((-\log P_\theta(y_i|\mathbf{x}_i, \mathcal{D}^{\mathcal{C}}) > R_{\mathcal{D}_{1:M}}(\theta))$ & $\rho_i \leq 1-\beta)$ **then**
6:          $\mathcal{D}_s \leftarrow \mathcal{D}_s \cup (\mathbf{x}_i, y_i)$;
7:      **end if**
8: **end for**
9: divide $\mathcal{D}_s$ into $\mathcal{D}_{s1}, \mathcal{D}_{s2}, \cdots, \mathcal{D}_{sK}$ with $\cup_{k=1}^K \mathcal{D}_{sk} = \mathcal{D}_s$;
10: **for** $k = 1, 2, \cdots, K$ **do**
11:      $\mathcal{D}_{1:M}^k = \mathcal{D}_{1:M} - \mathcal{D}_{sk}$;
12: **end for**
13: update parameters by optimizing $\theta^* = \arg\min_\theta w_0 R_{\mathcal{D}_{1:M}}(\theta) + \sum_{k=1}^K w_k R_{\mathcal{D}_{1:M}^k}(\theta)$;

**Output:** The learned optimal parameters $\theta^*$.

---

$\mathcal{D}_{s1}, \mathcal{D}_{s2}, \cdots, \mathcal{D}_{sK}$ which satisfy that $\cup_{k=1}^K \mathcal{D}_{sk} = \mathcal{D}_{1:M}$. Let $\mathcal{D}_{1:M}^0 = \mathcal{D}_{1:M} - \mathcal{D}_s$ and $\mathcal{D}_{1:M}^k = \mathcal{D}_{1:M} - \mathcal{D}_{sk}$ for all $k \in [K]$, then for any $\epsilon > 0$ and $1 > w_k > 0$ for all $k \in [K]$, $(w_0 + w_1 + \cdots + w_K = 1)$, $w_0 R_{\mathcal{D}_{1:M}}(\theta) + \sum_{k=1}^K w_k R_{\mathcal{D}_{1:M}^k}(\theta)$ and $w_0 R_{\mathcal{D}_{1:M}}(\theta) + (1 - w_0) R_{\mathcal{D}_{1:M}^0}(\theta)$ are $\delta_1$-stable and $\delta_2$-stable, respectively, then $\delta_1 \leq \delta_2$.

*Proof.* For simplicity, we denote $w_0 R_{\mathcal{D}_{1:M}}(\theta) + \sum_{k=1}^K w_k R_{\mathcal{D}_{1:M}^k}(\theta)$ as $R_1$ and $w_0 R_{\mathcal{D}_{1:M}}(\theta) + (1 - w_0) R_{\mathcal{D}_{1:M}^0}(\theta)$ as $R_2$. Let's assume that $R_\tau(\theta) - R_1 \in [-a_1, a_1]$ and $R_\tau(\theta) - R_2 \in [-a_2, a_2]$ are two random variables with 0 mean, where $a_1 = \sup\{R_\tau(\theta) - R_1\}$ and $a_2 = \sup\{R_\tau(\theta) - R_2\}$. Then, based on Markov's inequality and Hoeffding's lemma, we have

$$P(|R_\tau(\theta) - R_1| \leq \epsilon) \geq 1 - \frac{2e^{\frac{1}{2}t^2 a_1^2}}{e^{t\epsilon}}, \qquad P(|R_\tau(\theta) - R_2| \leq \epsilon) \geq 1 - \frac{2e^{\frac{1}{2}t^2 a_2^2}}{e^{t\epsilon}}. \tag{11}$$

$\forall k \in [K]$, $\mathcal{D}_{sk} \subset \mathcal{D}_s$ and $\forall (\mathbf{x}_i, y_i) \in \mathcal{D}_s$, $-\log P_\theta(y_i|\mathbf{x}_i, \mathcal{D}^{\mathcal{C}}) \leq R_{\mathcal{D}_{1:M}}(\theta)$, we have $R_{\mathcal{D}_{1:M}^k}(\theta) \leq R_{\mathcal{D}_{1:M}^0}(\theta)$. By combining the above inequalities over all $k \in [K]$, we have

$$\sum_{k=1}^K w_k R_{\mathcal{D}_{1:M}^k}(\theta) \leq \sum_{k=1}^K w_k R_{\mathcal{D}_{1:M}^0}(\theta) = (1 - w_0) R_{\mathcal{D}_{1:M}^0}(\mathcal{M}). \tag{12}$$

Thus, $\sup\{R_\tau(\theta) - R_1\} \leq \sup\{R_{1:M}(\theta) - R_2\}$, i.e. $a_1 \leq a_2$. Thus we have $\delta_1 \leq \delta_2$. $\qquad\square$

According to the above theorem, we can achieve model stability by selecting only one easily predicted subset.

## 6 Experiments

We started with learning predictive functions on synthetic datasets, and then high-dimensional tasks, e.g., system identification on physics engines, image completion, and Bayesian optimization, were performed to evaluate the properties of the NP-related models.

### 6.1 1D Regression

To verify the proposed stable solution, we combined the stable solution with different baseline NP classes (CNP [9], NP [10], ANP [14],ConvCNP [11], ConvNP [6], and their bootstrapping versions [18]) and compared them on 1D regression task. Among them, BCNP, BNP, BANP, BConvCNP, and NConvNP are recently proposed stable strategies for NPs with Bootstrap. Specifically, the stochastic process (SP) initializing with a 0 mean Gaussian Process (GP) $y^{(0)} \sim GP(0, k(\cdot, \cdot))$ indexed in the interval $x \in [-2.0, 2.0]$ were used to generate data, where the radial basis function kernel and Matern Kernel were adopted for model-data mismatch scenario. More detailed information can be obtained in the Appendix. We investigated the model performance in terms of different noise settings. We introduced Gaussian noise $\mathcal{N}(0,1)$ and added noise to different proportions of the data, such as $\{0\%, 5\%, 10\%, 15\%\}$. Table 1 lists the average log-likelihoods comparison in terms of different noise proportions. The best result is marked in bold. First, we can see that if we adopt the robust solution in baselines, the model achieves the best results on all the datasets, showing the effectiveness of the

Table 1: Average Log-likelihoods over all context and target points on realizations from Synthetic Stochastic Process on the different percent of added noise. Here we set the context size to 20. (Mean $\pm$ Std). Note that adding 'S' before the original model name is a model with our stable solution.

| Kernel | Method | Original | | Noise(+5%) | | Noise(+10%) | | Noise(+15%) | |
|---|---|---|---|---|---|---|---|---|---|
| | | context | target | context | target | context | target | context | target |
| RBF | CNP | $0.8724_{\pm0.008}$ | $0.4334_{\pm0.007}$ | $0.8522_{\pm0.005}$ | $0.4001_{\pm0.010}$ | $0.8014_{\pm0.006}$ | $0.3552_{\pm0.004}$ | $0.7152_{\pm0.006}$ | $0.2853_{\pm0.005}$ |
| | BCNP | $0.9042_{\pm0.009}$ | $0.4589_{\pm0.006}$ | $0.8774_{\pm0.006}$ | $0.4278_{\pm0.008}$ | $0.8316_{\pm0.006}$ | $0.3767_{\pm0.005}$ | $0.7487_{\pm0.007}$ | $0.3017_{\pm0.006}$ |
| | SCNP | $\mathbf{0.9255}_{\pm0.008}$ | $\mathbf{0.4733}_{\pm0.004}$ | $\mathbf{0.8935}_{\pm0.005}$ | $\mathbf{0.4478}_{\pm0.006}$ | $\mathbf{0.8517}_{\pm0.004}$ | $\mathbf{0.3986}_{\pm0.005}$ | $\mathbf{0.7621}_{\pm0.005}$ | $\mathbf{0.3279}_{\pm0.006}$ |
| | NP | $0.8215_{\pm0.004}$ | $0.3853_{\pm0.005}$ | $0.8011_{\pm0.004}$ | $0.3511_{\pm0.006}$ | $0.7611_{\pm0.005}$ | $0.3042_{\pm0.008}$ | $0.6722_{\pm0.005}$ | $0.2435_{\pm0.007}$ |
| | BNP | $0.8722_{\pm0.004}$ | $0.4211_{\pm0.004}$ | $0.8321_{\pm0.003}$ | $0.3876_{\pm0.004}$ | $0.7922_{\pm0.004}$ | $0.3389_{\pm0.005}$ | $0.7189_{\pm0.004}$ | $0.2776_{\pm0.007}$ |
| | SNP | $\mathbf{0.8955}_{\pm0.003}$ | $\mathbf{0.4356}_{\pm0.004}$ | $\mathbf{0.8567}_{\pm0.003}$ | $\mathbf{0.4046}_{\pm0.005}$ | $\mathbf{0.8165}_{\pm0.003}$ | $\mathbf{0.3568}_{\pm0.006}$ | $\mathbf{0.7356}_{\pm0.003}$ | $\mathbf{0.2955}_{\pm0.006}$ |
| | ANP | $1.2563_{\pm0.002}$ | $0.5763_{\pm0.004}$ | $1.2245_{\pm0.007}$ | $0.5347_{\pm0.006}$ | $0.1742_{\pm0.005}$ | $0.4871_{\pm0.007}$ | $0.9821_{\pm0.005}$ | $0.4151_{\pm0.004}$ |
| | BANP | $1.2722_{\pm0.004}$ | $0.5887_{\pm0.006}$ | $1.2411_{\pm0.005}$ | $0.5471_{\pm0.005}$ | $0.1886_{\pm0.006}$ | $0.4917_{\pm0.006}$ | $1.0642_{\pm0.006}$ | $0.4327_{\pm0.005}$ |
| | SANP | $\mathbf{1.2831}_{\pm0.000}$ | $\mathbf{0.5994}_{\pm0.004}$ | $\mathbf{1.2564}_{\pm0.004}$ | $\mathbf{0.5578}_{\pm0.004}$ | $\mathbf{1.2052}_{\pm0.004}$ | $\mathbf{0.5025}_{\pm0.004}$ | $\mathbf{1.1243}_{\pm0.005}$ | $\mathbf{0.4356}_{\pm0.006}$ |
| | ConvCNP | $1.2631_{\pm0.002}$ | $0.6421_{\pm0.002}$ | $1.2333_{\pm0.005}$ | $0.5415_{\pm0.005}$ | $0.1827_{\pm0.004}$ | $0.4936_{\pm0.005}$ | $1.0241_{\pm0.006}$ | $0.4262_{\pm0.005}$ |
| | BConvCNP | $1.2761_{\pm0.004}$ | $0.6531_{\pm0.005}$ | $1.2476_{\pm0.004}$ | $0.5533_{\pm0.004}$ | $0.1931_{\pm0.007}$ | $0.4986_{\pm0.005}$ | $1.0716_{\pm0.006}$ | $0.4396_{\pm0.005}$ |
| | SConvCNP | $\mathbf{1.3991}_{\pm0.001}$ | $\mathbf{0.6793}_{\pm0.004}$ | $\mathbf{1.2651}_{\pm0.003}$ | $\mathbf{0.5623}_{\pm0.005}$ | $\mathbf{1.2126}_{\pm0.004}$ | $\mathbf{0.5096}_{\pm0.005}$ | $\mathbf{1.1331}_{\pm0.006}$ | $\mathbf{0.4461}_{\pm0.005}$ |
| | ConvNP | $1.2874_{\pm0.003}$ | $0.6503_{\pm0.004}$ | $1.2371_{\pm0.006}$ | $0.5451_{\pm0.006}$ | $0.1865_{\pm0.004}$ | $0.4965_{\pm0.006}$ | $0.9915_{\pm0.005}$ | $0.4335_{\pm0.006}$ |
| | BConvNP | $1.2922_{\pm0.004}$ | $0.6627_{\pm0.006}$ | $1.2505_{\pm0.004}$ | $0.5583_{\pm0.004}$ | $0.1971_{\pm0.005}$ | $0.5025_{\pm0.007}$ | $1.0731_{\pm0.006}$ | $0.4436_{\pm0.004}$ |
| | SConvNP | $\mathbf{1.4036}_{\pm0.002}$ | $\mathbf{0.6831}_{\pm0.003}$ | $\mathbf{1.2671}_{\pm0.005}$ | $\mathbf{0.5675}_{\pm0.004}$ | $\mathbf{1.2188}_{\pm0.003}$ | $\mathbf{0.5157}_{\pm0.005}$ | $\mathbf{1.1389}_{\pm0.004}$ | $\mathbf{0.4505}_{\pm0.005}$ |
| Matern | CNP | $0.8531_{\pm0.005}$ | $0.2431_{\pm0.010}$ | $0.8231_{\pm0.005}$ | $0.2144_{\pm0.010}$ | $0.7761_{\pm0.008}$ | $0.1784_{\pm0.007}$ | $0.7052_{\pm0.005}$ | $0.1452_{\pm0.006}$ |
| | BCNP | $0.8778_{\pm0.005}$ | $0.2762_{\pm0.009}$ | $0.8487_{\pm0.006}$ | $0.2477_{\pm0.009}$ | $0.8015_{\pm0.007}$ | $0.2051_{\pm0.007}$ | $0.7378_{\pm0.005}$ | $0.1766_{\pm0.006}$ |
| | SCNP | $\mathbf{0.8963}_{\pm0.003}$ | $\mathbf{0.2953}_{\pm0.006}$ | $\mathbf{0.8689}_{\pm0.005}$ | $\mathbf{0.2658}_{\pm0.007}$ | $\mathbf{0.8268}_{\pm0.006}$ | $\mathbf{0.2258}_{\pm0.006}$ | $\mathbf{0.7567}_{\pm0.004}$ | $\mathbf{0.1936}_{\pm0.005}$ |
| | NP | $0.7643_{\pm0.015}$ | $0.2041_{\pm0.015}$ | $0.7342_{\pm0.002}$ | $0.1725_{\pm0.008}$ | $0.6892_{\pm0.004}$ | $0.1542_{\pm0.006}$ | $0.6235_{\pm0.008}$ | $0.1342_{\pm0.007}$ |
| | BNP | $0.8156_{\pm0.005}$ | $0.2689_{\pm0.007}$ | $0.7789_{\pm0.004}$ | $0.2215_{\pm0.005}$ | $0.7421_{\pm0.005}$ | $0.2117_{\pm0.007}$ | $0.6715_{\pm0.006}$ | $0.1828_{\pm0.006}$ |
| | SNP | $\mathbf{0.8368}_{\pm0.006}$ | $\mathbf{0.2844}_{\pm0.003}$ | $\mathbf{0.8036}_{\pm0.003}$ | $\mathbf{0.2483}_{\pm0.003}$ | $\mathbf{0.7635}_{\pm0.004}$ | $\mathbf{0.2325}_{\pm0.004}$ | $\mathbf{0.6973}_{\pm0.004}$ | $\mathbf{0.2016}_{\pm0.005}$ |
| | ANP | $1.2421_{\pm0.002}$ | $0.6366_{\pm0.004}$ | $1.2115_{\pm0.001}$ | $0.6001_{\pm0.008}$ | $1.1784_{\pm0.004}$ | $0.1622_{\pm0.006}$ | $1.1252_{\pm0.007}$ | $0.5274_{\pm0.008}$ |
| | BANP | $1.3456_{\pm0.003}$ | $0.6514_{\pm0.005}$ | $1.3125_{\pm0.005}$ | $0.6115_{\pm0.002}$ | $1.2672_{\pm0.004}$ | $0.1711_{\pm0.005}$ | $1.2236_{\pm0.006}$ | $0.5306_{\pm0.006}$ |
| | SANP | $\mathbf{1.3721}_{\pm0.002}$ | $\mathbf{0.6653}_{\pm0.004}$ | $\mathbf{1.3461}_{\pm0.003}$ | $\mathbf{0.6256}_{\pm0.004}$ | $\mathbf{1.3011}_{\pm0.003}$ | $\mathbf{0.1782}_{\pm0.004}$ | $\mathbf{1.2457}_{\pm0.005}$ | $\mathbf{0.5356}_{\pm0.002}$ |
| | ConvCNP | $1.2515_{\pm0.003}$ | $0.6418_{\pm0.004}$ | $1.2226_{\pm0.006}$ | $0.6085_{\pm0.005}$ | $1.1832_{\pm0.005}$ | $0.1871_{\pm0.007}$ | $1.1326_{\pm0.005}$ | $0.5351_{\pm0.004}$ |
| | BConvCNP | $1.3527_{\pm0.005}$ | $0.6616_{\pm0.006}$ | $1.3252_{\pm0.005}$ | $0.6235_{\pm0.007}$ | $1.2767_{\pm0.006}$ | $0.1952_{\pm0.005}$ | $1.1315_{\pm0.006}$ | $0.5417_{\pm0.006}$ |
| | SConvCNP | $\mathbf{1.3852}_{\pm0.003}$ | $\mathbf{0.6731}_{\pm0.004}$ | $\mathbf{1.3364}_{\pm0.004}$ | $\mathbf{0.6335}_{\pm0.005}$ | $\mathbf{1.2831}_{\pm0.003}$ | $\mathbf{0.2037}_{\pm0.005}$ | $\mathbf{1.1521}_{\pm0.005}$ | $\mathbf{0.5557}_{\pm0.004}$ |
| | ConvNP | $1.2746_{\pm0.002}$ | $0.6557_{\pm0.005}$ | $1.2345_{\pm0.003}$ | $0.6015_{\pm0.005}$ | $1.1865_{\pm0.006}$ | $0.1943_{\pm0.005}$ | $1.1358_{\pm0.005}$ | $0.5397_{\pm0.003}$ |
| | BConvNP | $1.3356_{\pm0.004}$ | $0.6787_{\pm0.006}$ | $1.3305_{\pm0.005}$ | $0.6383_{\pm0.006}$ | $1.2851_{\pm0.004}$ | $0.2015_{\pm0.005}$ | $1.1415_{\pm0.006}$ | $0.5338_{\pm0.003}$ |
| | SConvNP | $\mathbf{1.3878}_{\pm0.002}$ | $\mathbf{0.6836}_{\pm0.004}$ | $\mathbf{1.3435}_{\pm0.005}$ | $\mathbf{0.6417}_{\pm0.004}$ | $\mathbf{1.2866}_{\pm0.004}$ | $\mathbf{0.2025}_{\pm0.005}$ | $\mathbf{1.1521}_{\pm0.005}$ | $\mathbf{0.5363}_{\pm0.006}$ |

stable solution. Besides, performances on all methods become less accurate in more complicated settings, while our solution has fewer effects. One interesting observation is that the improvements against the base model on CNP are less significant than NP and ANP. The possible reason is that CNP only predicts points out of the context set.

To investigate the model's ability to address model-data mismatch scenarios, we conducted experiments on 1D regression tasks with Periodic kernel. Following the setting of BANP and similar noise settings of our previous kernels, we list the results on both original data and noise(+15) data in Table 2. In this model-data mismatch data, stable versions still significantly outperform their corresponding original versions.

Table 2: Experiments on 1D regression data with Periodic kernel.

| Periodic | Original | | Noise(+15%) | |
|---|---|---|---|---|
| | context | target | context | target |
| ANP | $0.5730_{\pm0.006}$ | $-4.2345_{\pm0.005}$ | $0.3521_{\pm0.005}$ | $-5.3211_{\pm0.007}$ |
| ConvCNP | $0.5983_{\pm0.006}$ | $-4.0215_{\pm0.005}$ | $0.3658_{\pm0.005}$ | $-4.5233_{\pm0.008}$ |
| ConvNP | $0.6125_{\pm0.005}$ | $-3.8952_{\pm0.006}$ | $0.3756_{\pm0.006}$ | $-4.3413_{\pm0.008}$ |
| BANP | $0.6253_{\pm0.003}$ | $-3.5413_{\pm0.005}$ | $0.3651_{\pm0.006}$ | $-4.2511_{\pm0.015}$ |
| BConvCNP | $0.6342_{\pm0.005}$ | $-3.4142_{\pm0.004}$ | $0.3712_{\pm0.004}$ | $-4.1750_{\pm0.008}$ |
| BConvNP | $0.6355_{\pm0.004}$ | $-3.3627_{\pm0.005}$ | $0.3768_{\pm0.008}$ | $-4.0116_{\pm0.007}$ |
| SANP | $0.6315_{\pm0.002}$ | $-3.3317_{\pm0.005}$ | $0.3748_{\pm0.004}$ | $-4.0515_{\pm0.003}$ |
| SConvCNP | $0.6433_{\pm0.002}$ | $-3.1515_{\pm0.004}$ | $0.3866_{\pm0.005}$ | $-3.9851_{\pm0.004}$ |
| SConvNP | $0.6551_{\pm0.000}$ | $-3.1062_{\pm0.004}$ | $0.3981_{\pm0.002}$ | $-3.8895_{\pm0.004}$ |

## 6.2 Image Completion

Image completion can be regarded as a 2D function regression task and be interpreted as being generated from a stochastic process (since there are dependencies between pixel values). Following the setting in previous work [14], we trained the NPs on EMNIST [4] and $32 \times 32$ CELEBA [23] using the standard train/test split with up to 200 context/target points at training. Detailed experiment settings are given in the Appendix. We evaluated average log-likelihoods over all points on realizations from image completion. Table 3 lists the comparisons between NPs with and without our stable solution in terms of original setting and noise setting, and the performance demonstrates the superiority of our stable solution.

## 6.3 System Identification on Physics Engines

The second synthetic experiment focuses on evaluating model dynamics on a classical simulator, Cart-Pole systems, which is detailed in [7, 27]. The Cart-Pole swing-up task is a standard benchmark for nonlinear control due to the non-linearity in the dynamics, and the requirement for nonlinear

Table 3: Average Log-likelihoods over all context and target points on EMNIST and CELEBA.

| Dataset | Method | Original | | Noise(+5%) | | Noise(+10%) | | Noise(+15%) | |
|---|---|---|---|---|---|---|---|---|---|
| | | context | target | context | target | context | target | context | target |
| EMNIST | CNP | $0.9522_{\pm0.023}$ | $0.7515_{\pm0.0015}$ | $0.8977_{\pm0.0016}$ | $0.6336_{\pm0.017}$ | $0.8242_{\pm0.0018}$ | $0.5784_{\pm0.009}$ | $0.6566_{\pm0.0017}$ | $0.5341_{\pm0.016}$ |
| | BCNP | $0.9678_{\pm0.010}$ | $0.8058_{\pm0.008}$ | $0.9015_{\pm0.008}$ | $0.6711_{\pm0.009}$ | $0.8415_{\pm0.007}$ | $0.6089_{\pm0.009}$ | $0.6788_{\pm0.006}$ | $0.5715_{\pm0.006}$ |
| | SCNP | $\mathbf{0.9716}_{\pm0.008}$ | $\mathbf{0.8343}_{\pm0.006}$ | $\mathbf{0.9251}_{\pm0.008}$ | $\mathbf{0.6971}_{\pm0.007}$ | $\mathbf{0.8674}_{\pm0.006}$ | $\mathbf{0.6343}_{\pm0.007}$ | $\mathbf{0.6986}_{\pm0.005}$ | $\mathbf{0.5877}_{\pm0.005}$ |
| | NP | $0.9678_{\pm0.004}$ | $0.7756_{\pm0.005}$ | $0.9011_{\pm0.009}$ | $0.6941_{\pm0.006}$ | $0.8544_{\pm0.009}$ | $0.6455_{\pm0.007}$ | $0.7034_{\pm0.009}$ | $0.5865_{\pm0.006}$ |
| | BNP | $0.9757_{\pm0.002}$ | $0.8358_{\pm0.005}$ | $0.8116_{\pm0.007}$ | $0.7625_{\pm0.006}$ | $0.8759_{\pm0.007}$ | $0.6773_{\pm0.007}$ | $0.7451_{\pm0.006}$ | $0.6237_{\pm0.005}$ |
| | SNP | $\mathbf{0.9847}_{\pm0.005}$ | $\mathbf{0.8562}_{\pm0.006}$ | $\mathbf{0.8368}_{\pm0.006}$ | $\mathbf{0.7844}_{\pm0.005}$ | $\mathbf{0.8984}_{\pm0.005}$ | $\mathbf{0.6984}_{\pm0.005}$ | $\mathbf{0.7653}_{\pm0.005}$ | $\mathbf{0.6456}_{\pm0.004}$ |
| | ANP | $1.1125_{\pm0.002}$ | $1.0321_{\pm0.004}$ | $0.9815_{\pm0.002}$ | $0.6366_{\pm0.006}$ | $0.9021_{\pm0.004}$ | $0.7053_{\pm0.008}$ | $0.8454_{\pm0.002}$ | $0.7034_{\pm0.005}$ |
| | BANP | $1.1355_{\pm0.003}$ | $1.0615_{\pm0.005}$ | $1.0236_{\pm0.002}$ | $0.6549_{\pm0.005}$ | $0.9155_{\pm0.004}$ | $0.7521_{\pm0.004}$ | $0.8612_{\pm0.003}$ | $0.7515_{\pm0.005}$ |
| | SANP | $\mathbf{1.1531}_{\pm0.000}$ | $\mathbf{1.0877}_{\pm0.004}$ | $\mathbf{1.0421}_{\pm0.002}$ | $\mathbf{0.6776}_{\pm0.005}$ | $\mathbf{0.9321}_{\pm0.002}$ | $\mathbf{0.7843}_{\pm0.006}$ | $\mathbf{0.8732}_{\pm0.003}$ | $\mathbf{0.7657}_{\pm0.005}$ |
| | ConvCNP | $1.1363_{\pm0.002}$ | $1.0461_{\pm0.004}$ | $1.0252_{\pm0.006}$ | $0.6448_{\pm0.005}$ | $0.9116_{\pm0.005}$ | $0.7115_{\pm0.006}$ | $0.8621_{\pm0.005}$ | $0.7246_{\pm0.004}$ |
| | BConvCNP | $1.1425_{\pm0.004}$ | $1.0787_{\pm0.006}$ | $1.0311_{\pm0.005}$ | $0.6626_{\pm0.005}$ | $0.9252_{\pm0.006}$ | $0.7617_{\pm0.006}$ | $0.8717_{\pm0.005}$ | $0.7627_{\pm0.005}$ |
| | SConvCNP | $\mathbf{1.1631}_{\pm0.003}$ | $\mathbf{1.0894}_{\pm0.004}$ | $\mathbf{1.0563}_{\pm0.004}$ | $\mathbf{0.6778}_{\pm0.004}$ | $\mathbf{0.9356}_{\pm0.005}$ | $\mathbf{0.7885}_{\pm0.005}$ | $\mathbf{0.8813}_{\pm0.005}$ | $\mathbf{0.7692}_{\pm0.006}$ |
| | ConvNP | $1.1415_{\pm0.002}$ | $1.0563_{\pm0.004}$ | $1.0286_{\pm0.006}$ | $0.6536_{\pm0.006}$ | $0.9168_{\pm0.005}$ | $0.7171_{\pm0.007}$ | $0.8675_{\pm0.005}$ | $0.7368_{\pm0.005}$ |
| | BConvNP | $1.1526_{\pm0.004}$ | $1.0837_{\pm0.005}$ | $1.0415_{\pm0.005}$ | $0.6684_{\pm0.005}$ | $0.9285_{\pm0.006}$ | $0.7762_{\pm0.006}$ | $0.8742_{\pm0.006}$ | $0.7727_{\pm0.005}$ |
| | SConvNP | $\mathbf{1.1753}_{\pm0.003}$ | $\mathbf{1.0934}_{\pm0.004}$ | $\mathbf{1.0641}_{\pm0.004}$ | $\mathbf{0.6837}_{\pm0.005}$ | $\mathbf{0.9402}_{\pm0.005}$ | $\mathbf{0.7925}_{\pm0.005}$ | $\mathbf{0.8923}_{\pm0.005}$ | $\mathbf{0.7853}_{\pm0.005}$ |
| CELEBA | CNP | $1.0323_{\pm0.016}$ | $0.7845_{\pm0.013}$ | $1.0177_{\pm0.016}$ | $0.7438_{\pm0.017}$ | $0.8956_{\pm0.009}$ | $0.7344_{\pm0.011}$ | $0.7677_{\pm0.012}$ | $0.6096_{\pm0.009}$ |
| | BCNP | $1.0452_{\pm0.009}$ | $0.8015_{\pm0.008}$ | $1.0275_{\pm0.009}$ | $0.7726_{\pm0.008}$ | $0.9351_{\pm0.006}$ | $0.8376_{\pm0.009}$ | $0.8015_{\pm0.010}$ | $0.6816_{\pm0.008}$ |
| | SCNP | $\mathbf{1.0525}_{\pm0.008}$ | $\mathbf{0.8243}_{\pm0.006}$ | $\mathbf{1.0348}_{\pm0.006}$ | $\mathbf{0.7868}_{\pm0.006}$ | $\mathbf{0.9562}_{\pm0.004}$ | $\mathbf{0.8545}_{\pm0.008}$ | $\mathbf{0.8344}_{\pm0.006}$ | $\mathbf{0.7045}_{\pm0.005}$ |
| | NP | $1.1333_{\pm0.004}$ | $0.8766_{\pm0.006}$ | $1.1043_{\pm0.015}$ | $0.8355_{\pm0.015}$ | $1.0034_{\pm0.008}$ | $0.8456_{\pm0.006}$ | $0.8935_{\pm0.006}$ | $0.6893_{\pm0.006}$ |
| | BNP | $1.1732_{\pm0.005}$ | $0.8901_{\pm0.006}$ | $1.1378_{\pm0.007}$ | $0.8678_{\pm0.006}$ | $1.0411_{\pm0.008}$ | $0.8711_{\pm0.005}$ | $0.9256_{\pm0.008}$ | $0.7671_{\pm0.006}$ |
| | SNP | $\mathbf{1.1952}_{\pm0.005}$ | $\mathbf{0.9062}_{\pm0.006}$ | $\mathbf{1.1565}_{\pm0.005}$ | $\mathbf{0.8846}_{\pm0.005}$ | $\mathbf{1.0542}_{\pm0.006}$ | $\mathbf{0.8956}_{\pm0.004}$ | $\mathbf{0.9425}_{\pm0.006}$ | $\mathbf{0.7985}_{\pm0.005}$ |
| | ANP | $1.1633_{\pm0.002}$ | $1.0163_{\pm0.004}$ | $1.1377_{\pm0.004}$ | $0.9866_{\pm0.006}$ | $1.0418_{\pm0.004}$ | $0.8845_{\pm0.006}$ | $0.9363_{\pm0.004}$ | $0.7346_{\pm0.008}$ |
| | BANP | $1.1751_{\pm0.002}$ | $1.0389_{\pm0.005}$ | $1.1488_{\pm0.004}$ | $1.0155_{\pm0.005}$ | $1.0602_{\pm0.005}$ | $0.9255_{\pm0.006}$ | $0.9489_{\pm0.004}$ | $0.8415_{\pm0.007}$ |
| | SANP | $\mathbf{1.1854}_{\pm0.000}$ | $\mathbf{1.0594}_{\pm0.004}$ | $\mathbf{1.1685}_{\pm0.002}$ | $\mathbf{1.0353}_{\pm0.004}$ | $\mathbf{1.0772}_{\pm0.002}$ | $\mathbf{0.9455}_{\pm0.004}$ | $\mathbf{0.9655}_{\pm0.003}$ | $\mathbf{0.8673}_{\pm0.005}$ |
| | ConvCNP | $1.1697_{\pm0.004}$ | $1.0366_{\pm0.004}$ | $1.1445_{\pm0.006}$ | $0.9947_{\pm0.006}$ | $1.0542_{\pm0.005}$ | $0.8971_{\pm0.007}$ | $0.9521_{\pm0.005}$ | $0.7563_{\pm0.006}$ |
| | BConvCNP | $1.1822_{\pm0.004}$ | $1.0467_{\pm0.004}$ | $1.1511_{\pm0.004}$ | $1.0271_{\pm0.005}$ | $1.0686_{\pm0.006}$ | $0.9317_{\pm0.006}$ | $0.9542_{\pm0.005}$ | $0.8527_{\pm0.005}$ |
| | SConvCNP | $\mathbf{1.1889}_{\pm0.003}$ | $\mathbf{1.0615}_{\pm0.005}$ | $\mathbf{1.1753}_{\pm0.004}$ | $\mathbf{1.0478}_{\pm0.004}$ | $\mathbf{1.0752}_{\pm0.004}$ | $\mathbf{0.9485}_{\pm0.005}$ | $\mathbf{0.9693}_{\pm0.007}$ | $\mathbf{0.8714}_{\pm0.006}$ |
| | ConvNP | $1.1767_{\pm0.004}$ | $1.0451_{\pm0.004}$ | $1.1485_{\pm0.005}$ | $1.0156_{\pm0.006}$ | $1.0594_{\pm0.005}$ | $0.9066_{\pm0.007}$ | $0.9573_{\pm0.004}$ | $0.7779_{\pm0.005}$ |
| | BConvNP | $1.1822_{\pm0.004}$ | $1.0555_{\pm0.004}$ | $1.1573_{\pm0.004}$ | $1.0351_{\pm0.005}$ | $1.0762_{\pm0.005}$ | $0.9461_{\pm0.003}$ | $0.9661_{\pm0.006}$ | $0.8636_{\pm0.003}$ |
| | SConvNP | $\mathbf{1.1867}_{\pm0.003}$ | $\mathbf{1.0668}_{\pm0.004}$ | $\mathbf{1.1846}_{\pm0.004}$ | $\mathbf{1.0487}_{\pm0.005}$ | $\mathbf{1.0863}_{\pm0.004}$ | $\mathbf{0.9511}_{\pm0.005}$ | $\mathbf{0.9743}_{\pm0.006}$ | $\mathbf{0.8836}_{\pm0.003}$ |

Table 4: Bayesian optimization experiments on data generated by different GP kernels.

| Method | ANP | BANP | SANP | ConvCNP | BConvCNP | SConvCNP | ConvNP | BConvNP | SConvNP |
|---|---|---|---|---|---|---|---|---|---|
| RBF | $0.1245_{\pm0.003}$ | $0.1341_{\pm0.003}$ | $\mathbf{0.1142}_{\pm0.002}$ | $0.1215_{\pm0.002}$ | $0.1168_{\pm0.003}$ | $\mathbf{0.1037}_{\pm0.002}$ | $0.1197_{\pm0.002}$ | $0.1156_{\pm0.003}$ | $\mathbf{0.1015}_{\pm0.003}$ |
| Matern | $0.1518_{\pm0.003}$ | $0.1316_{\pm0.004}$ | $\mathbf{0.1201}_{\pm0.002}$ | $0.1489_{\pm0.003}$ | $0.1301_{\pm0.002}$ | $\mathbf{0.1216}_{\pm0.002}$ | $0.1446_{\pm0.002}$ | $0.1242_{\pm0.003}$ | $\mathbf{0.1204}_{\pm0.004}$ |
| Periodic | $0.1892_{\pm0.002}$ | $0.1788_{\pm0.005}$ | $\mathbf{0.1672}_{\pm0.001}$ | $0.1652_{\pm0.002}$ | $0.1526_{\pm0.004}$ | $\mathbf{0.1487}_{\pm0.003}$ | $0.1611_{\pm0.002}$ | $0.1498_{\pm0.004}$ | $0.1446_{\pm0.002}$ |

controllers to successfully swing up and balance the pendulum. More detailed experimental settings are given in the Appendix.

For each configuration of the simulator including training and testing environments, we sampled 400 trajectories of the horizon as 10 steps using a random controller. During the testing process, 100 state transition pairs were randomly selected for each configuration of the environment, working as the maximum context points to identify the configuration of dynamics. Figure 2 shows the predictive Log-Likelihoods (LL) and Mean Average Error (MAE) on Cart-Pole State

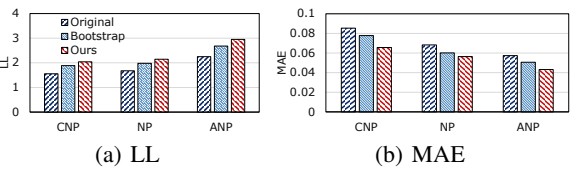

(a) LL          (b) MAE

Figure 2: The predictive Log Likelihood (LL) and Mean Average Error (MAE) on Cart-Pole state transition testing dataset.

Transition Testing Dataset. We can see that our stable NPs achieve better likelihood and lower prediction error than the original ones. The variances on the stable one are consistently smaller than all original baselines.

## 6.4 Bayesian Optimization

Following the setting in BANP [18], we conducted the Bayesian optimization experiment. Taking GP data with RBF, Matern, and Periodic prior functions as examples, we gave the results of ANP, ConvCNP, ConvNP, Boostraping versions, and our stable versions. To maintain consistent comparison, we standardized the initializations and normalized the results. We reported the best simple regret, which represents the difference between the current best observation and the global optimum. As shown in Table 4, we can see that our stable solutions consistently achieve lower regret than other NPs.

Table 5: Predator-prey model results.

| Method | ANP | BANP | SANP | ConvCNP | BConvCNP | SConvCNP | ConvNP | BConvNP | SConvNP |
|---|---|---|---|---|---|---|---|---|---|
| Simulated-context | $2.5801_{\pm 0.003}$ | $2.5912_{\pm 0.002}$ | $\mathbf{2.6127}_{\pm 0.004}$ | $2.5912_{\pm 0.002}$ | $2.6068_{\pm 0.003}$ | $\mathbf{2.6164}_{\pm 0.002}$ | $2.5925_{\pm 0.002}$ | $2.6125_{\pm 0.005}$ | $\mathbf{2.6231}_{\pm 0.003}$ |
| Simulated-target | $1.8265_{\pm 0.003}$ | $1.8635_{\pm 0.004}$ | $\mathbf{1.8844}_{\pm 0.002}$ | $1.8352_{\pm 0.004}$ | $1.8524_{\pm 0.003}$ | $\mathbf{1.9516}_{\pm 0.002}$ | $1.9228_{\pm 0.005}$ | $1.9442_{\pm 0.003}$ | $\mathbf{1.9662}_{\pm 0.004}$ |
| Real-context | $1.7234_{\pm 0.002}$ | $1.8496_{\pm 0.005}$ | $\mathbf{1.8412}_{\pm 0.001}$ | $1.7956_{\pm 0.002}$ | $1.8026_{\pm 0.004}$ | $\mathbf{1.8744}_{\pm 0.003}$ | $1.8342_{\pm 0.002}$ | $1.8476_{\pm 0.005}$ | $\mathbf{1.8796}_{\pm 0.002}$ |
| Real-target | $-7.8042_{\pm 0.002}$ | $-5.4836_{\pm 0.004}$ | $\mathbf{-5.2527}_{\pm 0.001}$ | $-5.3414_{\pm 0.005}$ | $-5.1526_{\pm 0.004}$ | $\mathbf{-5.3513}_{\pm 0.004}$ | $-5.3155_{\pm 0.002}$ | $-5.2615_{\pm 0.004}$ | $\mathbf{-5.2145}_{\pm 0.003}$ |

## 6.5 Predator-prey Models

Following [18] and [11], we consider the Lotka–Volterra model [30], which is used to describe the evolution of predator–prey populations. We first trained the models using simulated data generated from a Lotka-Volterra model and tested them on real-world data (Hudson's Bay hare-lynx data), which is quite different from the simulated data and can be considered as a mismatch scenario. Table 5 lists the results on both simulated and real data. Similar to the previous observation, our stable version still outperforms the original version. Among stable versions, SConvNP achieves the best performance.

## 6.6 Ablation Study

The key parameter in our stable solution is the number of hard predictable subsets $K$. Taking SANP as an example, we investigated the average log-likelihood in terms of different $K$ on the 1D regression task, as shown in Figure 3. We can see that SANP performs better as $K$ increases, reaches the best value at around $K = 4$, and then becomes stable in performance as $K$ grows larger. As proved in Theorem 5.3, optimization on $\mathcal{D}_{1:M}$ and more than one hard predictable subset of $\mathcal{D}_{1:M}$ can achieve more stable prediction. We also conducted different experiments to explore the impact of different $w_k$. Taking the 1D regression task with RBF-GP data as an example, we set $K = 3$ and different $w_k$ for experiments, as shown in Table 6. It can be seen from the table below that when different weights are set, the model using a stable strategy is better than the original model, and when the weight is set to be equal, its performance is optimal. In addition, when the value of $w_k(k \geq 1)$ is significantly different from $w_0$, such as $(0.625, 0.125, 0.125, 0.125)$ and $(0.0625, 0.3125, 0.3125, 0.3125)$, its performance is more significantly reduced compared to $(0.25, 0.25, 0.25, 0.25)$, but it still has a significant improvement compared to the original non-stable model.

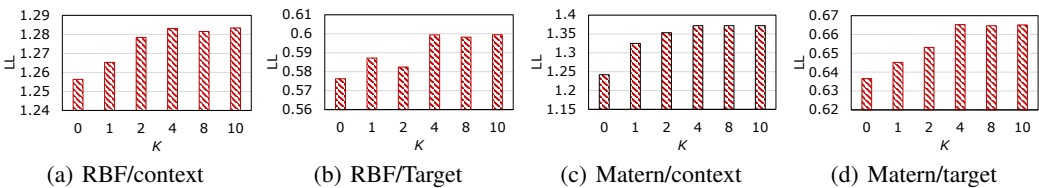

(a) RBF/context    (b) RBF/Target    (c) Matern/context    (d) Matern/target

Figure 3: Log-likelihood (SANP) comparisons with different $K$ on 1D regression task.

Table 6: Log-likelihood comparisons with different $w_k$ on 1D regression task.

| Weights | (0.25,0.25,0.25,0.25) | (0.4,0.2.0.2,0.2) | (0.625,0.125,0.125,0.125) | (0.1,0.3,0.3,0.3) | (0.0625,0.3125,0.3125,0.3125) |
|---|---|---|---|---|---|
| SCNP | (0.9255, 0.4125) | (0.9127, 0.4019) | (0.9035, 0.3998) | (0.9149, 0.4086) | (0.8927, 0.3991) |
| SNP | (0.8955, 0.3925) | (0.8737, 0.3817) | (0.8716, 0.3809) | (0.8831, 0.3859) | (0.8657, 0.3775) |
| SANP | (1.2831, 0.5215) | (1.2776, 0.5187) | (1.2738, 0.5196) | (1.2791, 0.5203) | (1.2712, 0.5193) |
| SConvCNP | (1.3991, 0.5996) | (1.3916, 0.5933) | (1.3841, 0.5915) | (1.3934, 0.5946) | (1.3854, 0.5919) |
| SConvNP | (1.4036, 0.6015) | (1.3991, 0.6004) | (1.3931, 0.5992) | (1.4012, 0.6015) | (1.3957, 0.5995) |

## 7 Conclusion and Future Work

In this paper, we provided theoretical guidelines for deriving stable solutions for NPs, which can obtain good generalization performance. Experiments demonstrated the proposed stable solution can help NPs to achieve more accurate and stable predictions. Although the theoretical analysis we give is based on regression models, it is still open to question whether this conclusion is appropriate for classification models. Therefore, we are interested in extending our theory, expecting it to apply to more different types of tasks.

## Acknowledgement

This work was partly supported by the National Natural Science Foundation of China under Grant 62176020; the National Key Research and Development Program (2020AAA0106800); the Joint Foundation of the Ministry of Education (8091B042235); the Beijing Natural Science Foundation under Grant L211016; the Fundamental Research Funds for the Central Universities (2019JBZ110); and Chinese Academy of Sciences (OEIP-O-202004).

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

## A   Inductive biases

Here, we revisit some properties, which would help us understand NPs. First, we give a concept of Permutation Invariant Function which is the basic property of stochastic process, e.g., NPs.

**Definition A.1.** (Permutation Invariant Function) A function $f(\cdot) : \times_i^N \mathbb{R}^D \to \mathbb{R}^d$ mapping a set of data points $\{\mathbf{x}_i\}_{i=1}^N$ is Permutation Invariant Function if

$$\mathbf{x} = [\mathbf{x}_1, \mathbf{x}_2, \cdots, \mathbf{x}_N] \to f = [f_1(\mathbf{x}_{\pi(1:N)}), f_2(\mathbf{x}_{\pi(1:N)}), \cdots, f_d(\mathbf{x}_{\pi(1:N)})], \tag{13}$$

where $x_i \in \mathbb{R}^D$ and the function output is a $d$ dimensional vector. Operation $\pi : [1, 2, \cdots, N] \to [\pi_1, \pi_2, \cdots, \pi_N]$ is a permutation set over the order of elements in the set.

**Definition A.2.** (Permutation Equivariant Function) A function $f(\cdot) : \times_i^N \mathbb{R}^D \to \mathbb{R}^N$ mapping a set of data points $\{\mathbf{x}_i\}_{i=1}^N$ is Permutation Invariant Function if

$$\mathbf{X}_\pi = [\mathbf{x}_{\pi_1}, \mathbf{x}_{\pi_2}, \cdots, \mathbf{x}_{\pi_N}] \to f_\pi = \pi \circ f(\mathbf{x}_{1:N}), \tag{14}$$

where the function output contains $N$ elements keeping the order of inputs.

Permutation Equivariant Function keeps the order of elements in the output consistent with that in the input under any permutation operation $\pi$. Permutation invariant functions are candidate functions for learning embeddings of a set or other order uncorrelated data structure $\{\mathbf{x}_i\}_{i=1}^N$, and the invariant property is easy to be verified. Here, we give a mean operation structure over the output

$$F(\mathbf{X}_{\pi(1:N)}) = \left( \frac{1}{N} \sum_{i=1}^N \phi_1(\mathbf{x}_i), \frac{1}{N} \sum_{i=1}^N \phi_2(\mathbf{x}_i), \cdots, \frac{1}{N} \sum_{i=1}^N \phi_D(\mathbf{x}_i) \right) \tag{15}$$

## B   Model Architecture

We show the architectural details of the CNP, NP, and ANP models used for the 1D and 2D function regression experiments. The neural process aims to learn a stochastic process (random function) mapping target features $\mathbf{x}_i$ to prediction $y_i$ given the context set $\mathcal{D}^{\mathcal{C}}$ as training data (a realization from the stochastic process), i.e., learning

$$\log P\left(\mathbf{y}^{\mathcal{T}}|\mathbf{X}^{\mathcal{T}}, \mathcal{D}^{\mathcal{C}}\right) = \log P\left(\mathbf{y}|\mathbf{X}, \mathcal{D}^{\mathcal{C}}\right) = \sum_{i=1}^N P\left(y_i|\mathbf{x}_i, \mathcal{D}^{\mathcal{C}}\right). \tag{16}$$

Conditional neural process (CNP) [9] describes $P\left(y_i|\mathbf{x}_i, \mathcal{D}^{\mathcal{C}}\right)$ with a deterministic neural network taking $\mathcal{D}^{\mathcal{C}}$ to output the parameters of $P\left(y_i|\mathbf{x}_i, \mathcal{D}^{\mathcal{C}}\right)$. CNP consists of an encoder $f_{\text{enc}}(\cdot)$, an aggregator $f_{\text{agg}}(\cdot)$ and a decoder $f_{\text{dec}}(\cdot)$; the encoder summarizes $\mathcal{D}^{\mathcal{C}}$ and $\mathbf{x}_i$ into latent representations $[\mathbf{r}_1, \cdots, \mathbf{r}_{|\mathcal{C}|}] \in \mathbb{R}^{|\mathcal{C}| \times d}$ via permutation-invariant neural network [31], where $d$ is the number of latent dimensions, and aggregator summarizes the encoded context features to a single representation $\mathbf{r}^{\mathcal{C}}$, and decoder takes as input the aggregated representations $\mathbf{r}^{\mathcal{C}}$ and $\mathbf{x}_i$ and output the single output-specific mean $\mu_i$ and variance $\sigma_i^2$ for the corresponding value of $y_i$.

$$\begin{aligned} \mathbf{r}_i &= f_{\text{enc}}\left(\mathbf{x}_i, y_i\right), \quad i \in \mathcal{C} \\ \mathbf{r}^{\mathcal{C}} &= \frac{1}{|\mathcal{C}|} \sum_{i \in \mathcal{C}} \mathbf{r}_i \\ \phi &= f_{\text{agg}}(\mathbf{r}^{\mathcal{C}}) \\ (\mu_i, \sigma_i) &= f_{\text{dec}}(\phi, \mathbf{x}_i), \quad p\left(y_i|\mathbf{x}_i, \mathcal{D}^{\mathcal{C}}\right) = \mathcal{N}\left(y_i; \mu_i, \sigma_i^2\right) \quad i \in \mathcal{T} \end{aligned} \tag{17}$$

where $f_{\text{enc}}(\cdot)$ and $f_{\text{dec}}(\cdot)$ are feed-forward neural networks. The decoder output $\mu_i$ and variance $\sigma_i^2$ are predicted mean and variance. We use *Gaussian* distribution $\mathcal{N}(y_i; \mu_i, \sigma_i^2)$ as predictive distribution. CNP is trained to maximize the expected likelihood $\mathbb{E}_{P(\mathcal{T})}[P\left(y_i|\mathbf{x}_i, \mathcal{D}^{\mathcal{C}}\right)]$.

Neural process [10] further models functional uncertainty using a global latent variable. Unlike CNP, which maps a context into a deterministic representation $\widetilde{\mathbf{r}}_i$, NP encoders a context into a *Gaussian* latent variable $\mathbf{z}$, giving additional stochasticity in function construction. Following [14], we consider

an NP with both a deterministic path and latent path, where the deterministic path models the overall skeleton of the function $\widetilde{\mathbf{r}}_i$, and the latent path models the functional uncertainty:

$$
\begin{aligned}
\mathbf{r}_i &= f_{\text{enc}}^{(1)}\left(\mathbf{x}_i, y_i\right), \quad i \in \mathcal{C} \\
\mathbf{r}^{\mathcal{C}} &= \frac{1}{|\mathcal{C}|} \sum_{i \in \mathcal{C}} \mathbf{r}_i \\
\phi &= f_{\text{agg}}(\mathbf{r}) \\
(\mu_z, \sigma_z) &= f_{\text{enc}}^{(2)}\left(\mathcal{D}^{\mathcal{C}}\right), \quad q(\mathbf{z}|\mathcal{D}^{\mathcal{C}}) = \mathcal{N}(\mathbf{z}; \mu_z, \sigma_z^2) \\
(\mu_i, \sigma_i) &= f_{\text{dec}}(\phi, \mathbf{z}, \mathbf{x}_i), \quad p\left(y_i|\mathbf{x}_i, \mathbf{z}, \mathcal{D}^{\mathcal{C}}\right) = \mathcal{N}\left(y_i; \mu_i, \sigma_i^2\right) \quad i \in \mathcal{T}
\end{aligned}
\tag{18}
$$

with $f_{\text{enc}}^{(1)}(\cdot)$ and $f_{\text{enc}}^{(2)}(\cdot)$ having the same structure as $f_{\text{enc}}(\cdot)$ in Eq.(17). In this scenario, the conditional distribution is lower bounded as:

$$
\log P\left(\mathbf{y}|\mathbf{X}, \mathcal{D}^{\mathcal{C}}\right) \geq \sum_{i=1}^{N} \mathbb{E}_{q(\mathbf{z}|\mathcal{D}^{\mathcal{C}})}\left[\log \frac{P\left(y_i|\mathbf{x}_i, \mathbf{z}, \mathcal{D}^{\mathcal{C}}\right) P(\mathbf{z}|\mathcal{D}^{\mathcal{C}})}{q(\mathbf{z}|\mathbf{X}, \mathbf{y})}\right].
\tag{19}
$$

We further approximate $q(\mathbf{z}|\mathcal{D}^{\mathcal{C}}) \approx P(\mathbf{z}|\mathcal{D}^{\mathcal{C}})$ and train the model by maximizing this expected lower bound over tasks. Furthermore, ANP introduces attention mechanisms into NP to resolve the issue of under-fitting.

The architectural details of the CNP, NP, and ANP are the same as in [14]. Here we give the detailed architectures of the encoder and decoder of NPs.

## B.1  Encoder without attention

Encoder focuses on learning embeddings for each data point in the context set, and the basic component is multi-layer perceptron, which is defined by

$$
\text{MLP}(l, d_{in}, d_h, d_{out}) = \text{LINEAR}(d_h, d_{out}) \circ \underbrace{\left(\text{RELU} \circ \text{LINEAR}(d_h, d_h) \circ \cdots\right)}_{\times (l-2)} \circ \text{LINEAR}(d_h, d_{in})
\tag{20}
$$

where $l$ is the number of layers, $d_{in}$, $d_h$ and $d_{out}$ are dimensinalities of inputs, hidden unites and outputs. Here $\text{RELU}(\cdot)$ is adapted as activation function.

The encoder in Vanilla CNP uses a deterministic encoder which focuses on learning embeddings for each data point in context set.

$$
\begin{aligned}
\mathbf{r}_i &= \text{MLP}(l_{e1}, d_x + d_y, d_h, d_h)([\mathbf{x}_i, y_i]), \\
\mathbf{r}^{\mathcal{C}} &= \sum_{i \in \mathcal{C}} \mathbf{r}_i, \quad \phi = \text{MLP}(l_{e2}, d_h, d_h)(\mathbf{r}^{\mathcal{C}})
\end{aligned}
\tag{21}
$$

where $d_x$ and $d_y$ are the dimensionalities of $\mathbf{x}_i$ and $y_i$.

To follow the encoder structure in NP, we introduce another encoder aligned with original deterministic encoder to permit the same number parameters, i.e.,

$$
\begin{aligned}
\mathbf{r}_i^{(1)} &= \text{MLP}(l_{e1}, d_x + d_y, d_h, d_h)([\mathbf{x}_i, y_i]) \\
\mathbf{r}_{\mathcal{C}}^{(1)} &= \sum_{i \in \mathcal{C}} \mathbf{r}_i^{(1)}, \quad \phi_1 = \text{MLP}(l_{e2}, d_h, d_h)(\mathbf{r}_{\mathcal{C}}^{(1)}) \\
\mathbf{r}_i^{(2)} &= \text{MLP}(l_{e1}, d_x + d_y, d_h, d_h)([\mathbf{x}_i, y_i]) \\
\mathbf{r}_{\mathcal{C}}^{(2)} &= \sum_{i \in \mathcal{C}} \mathbf{r}_i^{(2)}, \quad \phi_2 = \text{MLP}(l_{e2}, d_h, d_h)(\mathbf{r}_{\mathcal{C}}^{(2)}) \\
\phi &= [\phi_1, \phi_2]
\end{aligned}
\tag{22}
$$

The encoder in NP contains a deterministic path and a latent path, i.e.,

$$\mathbf{r}_i^{(1)} = \text{MLP}(l_{de1}, d_x + d_y, d_h, d_h)([\mathbf{x}_i, y_i])$$
$$\mathbf{r}_\mathcal{C}^{(1)} = \sum_{i \in \mathcal{C}} \mathbf{r}_i^{(1)}, \quad \phi = \text{MLP}(l_{de2}, d_h, d_h)(\mathbf{r}_\mathcal{C}^{(1)})$$
$$\mathbf{r}_i^{(2)} = \text{MLP}(l_{la1}, d_x + d_y, d_h, d_h)([\mathbf{x}_i, y_i]) \tag{23}$$
$$\mathbf{r}_\mathcal{C}^{(2)} = \sum_{i \in \mathcal{C}} \mathbf{r}_i^{(2)}, \quad [\mu_z, \sigma_z'] = \text{MLP}(l_{la2}, d_h, d_h)(\mathbf{r}_\mathcal{C}^{(2)})$$
$$\sigma_z = 0.1 + 0.9 \cdot \text{SIGMOID}(\sigma_z'), \quad \mathbf{z} \sim \mathcal{N}(\mu_z, \text{diag}(\sigma_z^2)).$$

In this case, the encoder outputs deterministic representation $\phi$ and latent representation $\mathbf{z}$.

## B.2 Encoder with attention

The attention mechanism is widely used in NPs, Specifically, multi-head attention [26] is adapted, which is defined by

$$\mathbf{Q}' = \{\text{LINEAR}(d_q, d_{out})(\mathbf{q})\}_{\mathbf{q} \in \mathbf{Q}}, \quad \{\mathbf{Q}_i'\}_{i=1}^{n_{head}} = \text{SPLIT}(\mathbf{Q}', n_{head}),$$
$$\mathbf{K}' = \{\text{LINEAR}(d_k, d_{out})(\mathbf{k})\}_{\mathbf{k} \in \mathbf{K}}, \quad \{\mathbf{K}_i'\}_{i=1}^{n_{head}} = \text{SPLIT}(\mathbf{K}', n_{head}),$$
$$\mathbf{V}' = \{\text{LINEAR}(d_v, d_{out})(\mathbf{v})\}_{\mathbf{v} \in \mathbf{V}}, \quad \{\mathbf{V}_i'\}_{i=1}^{n_{head}} = \text{SPLIT}(\mathbf{V}', n_{head}),$$
$$\mathbf{H}_i = \text{SOFTMAX}\left(\mathbf{Q}_i'(\mathbf{K}_i')^\top / \sqrt{d_{out}}\right)\mathbf{V}_i', \quad \mathbf{H} = \text{CONCAT}\left(\{\mathbf{H}_i\}_{i=1}^{n_{head}}\right)$$
$$\mathbf{H}' = \text{LAYERNORM}(\mathbf{Q}' + \mathbf{H})$$
$$\text{MHA}(d_{out})(\mathbf{Q}, \mathbf{K}, \mathbf{V}) = \text{LAYERNORM}(\mathbf{H}' + \text{RELU}(\text{LINEAR}(d_{out}, d_{out})))$$
$$\tag{24}$$

where $d_q, d_v, d_k$ are the dimensionalities of query $\mathbf{Q}$, key $\mathbf{K}$, and value $\mathbf{V}$, respectively. $n_{head}$ is the number of head. Here Layer normalization [1] $\text{LAYERNORM}(\cdot)$ is adapted. It is easy to derive self-attention by setting $\mathbf{Q} = \mathbf{K} = \mathbf{V}$, i.e.,

$$\text{SA}(d_{out}))(\mathbf{X}) = \text{MHA}(d_{out})(\mathbf{X}, \mathbf{X}, \mathbf{X}) \tag{25}$$

For CNP, the encoder with attention still contains two deterministic paths,

$$f_{qk} = \text{MLP}(l_{qk}, d_x, d_h, d_h)$$
$$\mathbf{Q} = f_{qk}(\mathbf{x}_i), \quad i \in \mathcal{T}$$
$$\mathbf{K} = \{f_{qk}(\mathbf{x}_i)\}, \quad i \in \mathcal{C}$$
$$\mathbf{V} = \text{SA}(d_h).(\{\text{MLP}(l_v, d_x + d_y, d_h, d_h)([\mathbf{x}_i, y_i])\}_{i \in \mathcal{C}})$$
$$\phi_1 = \text{MHA}(d_h)(\mathbf{Q}, \mathbf{K}, \mathbf{V}) \tag{26}$$
$$\mathbf{H} = \text{SA}(d_h)(\{\text{RELU} \circ \text{MLP}(l_{e1}, d_x + d_y, d_h, d_h)([\mathbf{x}_i, y_i])\}_{i \in \mathcal{C}})$$
$$\phi_2 = \text{MLP}(l_e, d_h, d_h)\left(\frac{1}{|\mathcal{C}|}\sum_{i \in \mathcal{C}} \mathbf{h}_i\right)$$
$$\phi = [\phi_1, \phi_2]$$

Similarly, encoder with attention in NP contains a deterministic path and a latent path, i.e.,

$$f_{qk} = \text{MLP}(l_{qk}, d_x, d_h, d_h)$$
$$\mathbf{Q} = f_{qk}(\mathbf{x}_i), \quad i \in \mathcal{T}$$
$$\mathbf{K} = \{f_{qk}(\mathbf{x}_i)\}, \quad i \in \mathcal{C} \tag{27}$$
$$\mathbf{V} = \text{SA}(d_h).(\{\text{MLP}(l_v, d_x + d_y, d_h, d_h)([\mathbf{x}_i, y_i])\}_{i \in \mathcal{C}})$$
$$\phi = \text{MHA}(d_h)(\mathbf{Q}, \mathbf{K}, \mathbf{V})$$

and

$$\mathbf{H} = \mathrm{SA}(d_h) \left( \{\mathrm{RELU} \circ \mathrm{MLP}(l_{e1}, d_x + d_y, d_h, d_h)([\mathbf{x}_i, y_i])\}_{i \in \mathcal{C}} \right)$$

$$[\mu_z, \sigma_z'] = \mathrm{MLP}(l_{la}, d_h, d_h) \left( \frac{1}{|\mathcal{C}|} \sum_{i \in \mathcal{C}} \mathbf{h}_i \right)$$

$$\sigma_z = 0.1 + 0.9 \cdot \mathrm{SIGMOID}(\sigma_z'),$$

$$\mathbf{z} \sim \mathcal{N}(\mu_z, \mathrm{diag}(\sigma_z^2)). \tag{28}$$

### B.3 Decoder

The decoder focuses on predicting output for target points based on the encoder's outputs $\phi$. For target point $\{\mathbf{x}_i\}_{i \in \mathcal{T}}$, the decoder of CNP is defined by

$$[\mu_i, \sigma_i'] = \mathrm{MLP}(d_{dec}, 2d_h + d_x, d_h, 2d_y)[\phi, \mathbf{x}_i], \quad i \in \mathcal{T}$$

$$\sigma_i = 0.1 + 0.9 \cdot \mathrm{SOFTPLUS}(\sigma_i') \tag{29}$$

$$y_i \sim \mathcal{N}(\mu_i, \sigma_i)$$

Decoder of NP is defined by

$$[\mu_i, \sigma_i'] = \mathrm{MLP}(d_{dec}, d_h + d_z + d_x, d_h, 2d_y)[\phi, \mathbf{x}_i, \mathbf{z}], \quad i \in \mathcal{T}$$

$$\sigma_i = 0.1 + 0.9 \cdot \mathrm{SOFTPLUS}(\sigma_i') \tag{30}$$

$$y_i \sim \mathcal{N}(\mu_i, \sigma_i)$$

## C  Implementation Details and Experiments

For CNP [9], BCNP [18], and our SCNP, we apply the encoder with attention described in Eq (26) and decoder described in Eq (29). For NP [10], ANP [14], BNP [18], BANP [18] and our SNP and SANP models, we apply encoder with attention described in Eq (27) and (28), and decoder described in Eq (30).

### C.1  1D Regression

For synthetic 1D regression experiments, the neural architectures for CNP, NP, ANP, BCNP, BNP, BANP, and our SCNP/SNP/SANP refer to Appendix B. The number of hidden units is $d_h = 128$ and latent representation $d_z = 128$. The number of layers are $l_e = l_{de} = l_{la} = l_{qk} = l_v = 2$.

We generate datasets for synthetic 1D regression. Specifically, the stochastic process (SP) initializes with a 0 mean Gaussian Process (GP) $y^{(0)} \sim GP(0, k(\cdot, \cdot))$ indexed in the interval $x \in [-2.0, 2.0]$, where the radial basis function kernel $k(x, x') = \sigma^2 \exp(-\|x - x'\|^2 / 2l^2)$ with $s \sim U(0.1, 1.0)$ and $\sigma \sim U(0.1, 0.6)$. Furthermore, GP with Matern Kernel is adopted for model-data mismatch scenario, which is defined as $k(x, x') = \sigma^2 (1 + \sqrt{5}d/l + 5d^2/(3l^2)) \exp(-\sqrt{5}d/l)$ and $d = \|x - x'\|$ with $s \sim U(0.1, 1.0)$ and $\sigma \sim U(0.1, 0.6)$. For a fair comparison, we set the same data generation, training, and testing for all models.

We trained all models for $100,000$ steps with each step computing updates with a batch containing 100 tasks. We used the Adam optimizer with an initial learning rate $5 \cdot 10^{-4}$ and decayed the learning rate using Cosine annealing scheme for baselines. For SCNP/SNP/SANP, we set $K = 3$. The size of the context $\mathcal{C}$ was drawn as $|\mathcal{C}| \sim U(3, 200)$. Testings were done for $3,000$ batches with each batch containing 16 tasks ($48,000$ tasks in total).

We investigate the model stability from the size of the context set and the percent of added noise. First, we conduct experiments on different size of context set, i.e., $|\mathcal{C}| \in \{20, 50, 100, 200\}$. Table 7 shows the Average Log-likelihoods performance comparison between different methods in terms of different context size. We can see that the performance becomes better with the increasing of $|\mathcal{C}|$ and NPs with stable solution still achieve better performance. Second, we investigate the model performance in terms of different noise setting. Here we introduce Gaussian noise $\mathcal{N}(0, 1)$ and add noise to different proportions of the data, such as $\{0\%, 5\%, 10\%, 15\%\}$. Table 8 lists the Average Log-likelihoods performance comparison in terms of different noise proportions.

Table 7: Average Log-likelihoods over all context and target points on realizations from Synthetic Stochastic Process on different size of context set. (Mean $\pm$ Std).

| Kernel | Method | 20 | | 50 | | 100 | | 200 | |
|---|---|---|---|---|---|---|---|---|---|
| | | context | target | context | target | context | target | context | target |
| RBF | CNP | $0.8724_{\pm0.008}$ | $0.4334_{\pm0.007}$ | $0.9533_{\pm0.005}$ | $0.4854_{\pm0.010}$ | $1.1224_{\pm0.005}$ | $0.5322_{\pm0.007}$ | $0.1563_{\pm0.004}$ | $0.5783_{\pm0.004}$ |
| | BCNP | $0.9015_{\pm0.009}$ | $0.4579_{\pm0.007}$ | $0.9787_{\pm0.007}$ | $0.5215_{\pm0.008}$ | $1.1687_{\pm0.007}$ | $0.5716_{\pm0.009}$ | $0.1985_{\pm0.005}$ | $0.6086_{\pm0.005}$ |
| | SCNP | $\mathbf{0.9255}_{\pm0.008}$ | $\mathbf{0.4733}_{\pm0.004}$ | $\mathbf{0.9944}_{\pm0.005}$ | $\mathbf{0.5433}_{\pm0.006}$ | $\mathbf{1.1833}_{\pm0.006}$ | $\mathbf{0.5918}_{\pm0.005}$ | $\mathbf{0.2111}_{\pm0.004}$ | $\mathbf{0.6333}_{\pm0.004}$ |
| | NP | $0.8215_{\pm0.004}$ | $0.3853_{\pm0.005}$ | $0.9124_{\pm0.006}$ | $0.4234_{\pm0.003}$ | $1.0855_{\pm0.003}$ | $0.4767_{\pm0.005}$ | $1.1225_{\pm0.002}$ | $0.5233_{\pm0.004}$ |
| | BNP | $0.8714_{\pm0.004}$ | $0.4122_{\pm0.004}$ | $0.9712_{\pm0.005}$ | $0.4718_{\pm0.004}$ | $1.1426_{\pm0.005}$ | $0.5269_{\pm0.006}$ | $1.1716_{\pm0.006}$ | $0.5698_{\pm0.004}$ |
| | SNP | $\mathbf{0.8955}_{\pm0.003}$ | $\mathbf{0.4356}_{\pm0.004}$ | $\mathbf{0.9866}_{\pm0.004}$ | $\mathbf{0.4934}_{\pm0.005}$ | $\mathbf{1.1637}_{\pm0.004}$ | $\mathbf{0.5434}_{\pm0.004}$ | $\mathbf{1.1958}_{\pm0.003}$ | $\mathbf{0.5933}_{\pm0.003}$ |
| | ANP | $1.2563_{\pm0.002}$ | $0.5763_{\pm0.004}$ | $1.3233_{\pm0.002}$ | $0.6322_{\pm0.004}$ | $1.4633_{\pm0.003}$ | $0.6866_{\pm0.006}$ | $1.4982_{\pm0.006}$ | $0.7322_{\pm0.004}$ |
| | BANP | $1.2715_{\pm0.003}$ | $0.5878_{\pm0.004}$ | $1.3325_{\pm0.002}$ | $0.6465_{\pm0.004}$ | $1.4778_{\pm0.003}$ | $0.6915_{\pm0.004}$ | $1.5115_{\pm0.003}$ | $0.7436_{\pm0.005}$ |
| | SANP | $\mathbf{1.2831}_{\pm0.000}$ | $\mathbf{0.5994}_{\pm0.004}$ | $\mathbf{1.3452}_{\pm0.002}$ | $\mathbf{0.6577}_{\pm0.003}$ | $\mathbf{1.4898}_{\pm0.001}$ | $\mathbf{0.7043}_{\pm0.002}$ | $\mathbf{1.5285}_{\pm0.002}$ | $\mathbf{0.7534}_{\pm0.005}$ |
| Matern | CNP | $0.8531_{\pm0.006}$ | $0.2431_{\pm0.010}$ | $0.9123_{\pm0.006}$ | $0.2984_{\pm0.005}$ | $1.0522_{\pm0.004}$ | $0.3542_{\pm0.004}$ | $1.0984_{\pm0.008}$ | $0.4022_{\pm0.005}$ |
| | BCNP | $0.8765_{\pm0.006}$ | $0.2788_{\pm0.009}$ | $0.9411_{\pm0.006}$ | $0.3266_{\pm0.005}$ | $1.0752_{\pm0.004}$ | $0.3762_{\pm0.004}$ | $1.1245_{\pm0.007}$ | $0.4326_{\pm0.006}$ |
| | SCNP | $\mathbf{0.8963}_{\pm0.003}$ | $\mathbf{0.2953}_{\pm0.006}$ | $\mathbf{0.9555}_{\pm0.004}$ | $\mathbf{0.3467}_{\pm0.003}$ | $\mathbf{1.0967}_{\pm0.003}$ | $\mathbf{0.3967}_{\pm0.003}$ | $\mathbf{1.1467}_{\pm0.008}$ | $\mathbf{0.4556}_{\pm0.005}$ |
| | NP | $0.7643_{\pm0.015}$ | $0.2041_{\pm0.015}$ | $0.8221_{\pm0.004}$ | $0.2547_{\pm0.004}$ | $0.9322_{\pm0.003}$ | $0.3155_{\pm0.004}$ | $1.0452_{\pm0.005}$ | $0.3563_{\pm0.004}$ |
| | BNP | $0.8052_{\pm0.008}$ | $0.2651_{\pm0.007}$ | $0.8678_{\pm0.003}$ | $0.3163_{\pm0.004}$ | $0.9672_{\pm0.003}$ | $0.3656_{\pm0.004}$ | $1.1052_{\pm0.005}$ | $0.4015_{\pm0.005}$ |
| | SNP | $\mathbf{0.8368}_{\pm0.006}$ | $\mathbf{0.2844}_{\pm0.005}$ | $\mathbf{0.8956}_{\pm0.002}$ | $\mathbf{0.3326}_{\pm0.004}$ | $\mathbf{0.9959}_{\pm0.003}$ | $\mathbf{0.3849}_{\pm0.004}$ | $\mathbf{1.1215}_{\pm0.003}$ | $\mathbf{0.4313}_{\pm0.004}$ |
| | ANP | $1.2421_{\pm0.002}$ | $0.6366_{\pm0.004}$ | $1.3022_{\pm0.001}$ | $0.6881_{\pm0.004}$ | $1.4211_{\pm0.004}$ | $0.7331_{\pm0.003}$ | $1.4631_{\pm0.002}$ | $0.7753_{\pm0.008}$ |
| | BANP | $1.3452_{\pm0.007}$ | $0.6513_{\pm0.004}$ | $1.4056_{\pm0.003}$ | $0.7015_{\pm0.004}$ | $1.4505_{\pm0.004}$ | $0.7531_{\pm0.003}$ | $1.4986_{\pm0.005}$ | $0.7996_{\pm0.006}$ |
| | SANP | $\mathbf{1.3721}_{\pm0.002}$ | $\mathbf{0.6653}_{\pm0.004}$ | $\mathbf{1.4322}_{\pm0.003}$ | $\mathbf{0.7126}_{\pm0.004}$ | $\mathbf{1.4633}_{\pm0.004}$ | $\mathbf{0.7644}_{\pm0.003}$ | $\mathbf{1.5153}_{\pm0.005}$ | $\mathbf{0.8125}_{\pm0.004}$ |

Table 8: Average Log-likelihoods over all context and target points on realizations from Synthetic Stochastic Process on different percent of added noise. Here we set the context size to 20. (Mean $\pm$ Std). Note that adding 'S' before the original model name is a model with our stable solution.

| Kernel | Method | Original | | Noise(+5%) | | Noise(+10%) | | Noise(+15%) | |
|---|---|---|---|---|---|---|---|---|---|
| | | context | target | context | target | context | target | context | target |
| RBF | CNP | $0.8724_{\pm0.008}$ | $0.4334_{\pm0.007}$ | $0.8522_{\pm0.005}$ | $0.4001_{\pm0.010}$ | $0.8014_{\pm0.006}$ | $0.3552_{\pm0.004}$ | $0.7152_{\pm0.006}$ | $0.2853_{\pm0.005}$ |
| | BCNP | $0.9042_{\pm0.009}$ | $0.4589_{\pm0.006}$ | $0.8774_{\pm0.006}$ | $0.4278_{\pm0.008}$ | $0.8316_{\pm0.006}$ | $0.3767_{\pm0.005}$ | $0.7487_{\pm0.007}$ | $0.3017_{\pm0.006}$ |
| | SCNP | $\mathbf{0.9255}_{\pm0.008}$ | $\mathbf{0.4733}_{\pm0.004}$ | $\mathbf{0.8935}_{\pm0.005}$ | $\mathbf{0.4478}_{\pm0.006}$ | $\mathbf{0.8517}_{\pm0.004}$ | $\mathbf{0.3986}_{\pm0.005}$ | $\mathbf{0.7621}_{\pm0.005}$ | $\mathbf{0.3279}_{\pm0.006}$ |
| | NP | $0.8215_{\pm0.004}$ | $0.3853_{\pm0.005}$ | $0.8011_{\pm0.004}$ | $0.3511_{\pm0.006}$ | $0.7611_{\pm0.005}$ | $0.3042_{\pm0.008}$ | $0.6722_{\pm0.005}$ | $0.2435_{\pm0.007}$ |
| | BNP | $8722_{\pm0.004}$ | $0.4211_{\pm0.004}$ | $0.8321_{\pm0.003}$ | $0.3876_{\pm0.004}$ | $0.7922_{\pm0.004}$ | $0.3389_{\pm0.005}$ | $0.7189_{\pm0.004}$ | $0.2776_{\pm0.007}$ |
| | SNP | $\mathbf{0.8955}_{\pm0.003}$ | $\mathbf{0.4356}_{\pm0.004}$ | $\mathbf{0.8567}_{\pm0.003}$ | $\mathbf{0.4046}_{\pm0.005}$ | $\mathbf{0.8165}_{\pm0.004}$ | $\mathbf{0.3568}_{\pm0.006}$ | $\mathbf{0.7356}_{\pm0.005}$ | $\mathbf{0.2955}_{\pm0.006}$ |
| | ANP | $1.2563_{\pm0.002}$ | $0.5763_{\pm0.004}$ | $1.2245_{\pm0.007}$ | $0.5347_{\pm0.006}$ | $0.1742_{\pm0.005}$ | $0.4871_{\pm0.007}$ | $0.9821_{\pm0.005}$ | $0.4151_{\pm0.004}$ |
| | BANP | $1.2722_{\pm0.004}$ | $0.5887_{\pm0.006}$ | $1.2411_{\pm0.005}$ | $0.5471_{\pm0.005}$ | $0.1886_{\pm0.006}$ | $0.4917_{\pm0.006}$ | $1.0642_{\pm0.006}$ | $0.4327_{\pm0.005}$ |
| | SANP | $\mathbf{1.2831}_{\pm0.000}$ | $\mathbf{0.5994}_{\pm0.004}$ | $\mathbf{1.2564}_{\pm0.004}$ | $\mathbf{0.5578}_{\pm0.004}$ | $\mathbf{1.2052}_{\pm0.004}$ | $\mathbf{0.5025}_{\pm0.006}$ | $\mathbf{1.1243}_{\pm0.005}$ | $\mathbf{0.4356}_{\pm0.006}$ |
| Matern | CNP | $0.8531_{\pm0.005}$ | $0.2431_{\pm0.010}$ | $0.8231_{\pm0.005}$ | $0.2144_{\pm0.010}$ | $0.7761_{\pm0.008}$ | $0.1784_{\pm0.007}$ | $0.7052_{\pm0.005}$ | $0.1452_{\pm0.006}$ |
| | BCNP | $0.8778_{\pm0.005}$ | $0.2762_{\pm0.009}$ | $0.8487_{\pm0.006}$ | $0.2477_{\pm0.009}$ | $0.8015_{\pm0.007}$ | $0.2051_{\pm0.007}$ | $0.7378_{\pm0.005}$ | $0.1766_{\pm0.006}$ |
| | SCNP | $\mathbf{0.8963}_{\pm0.003}$ | $\mathbf{0.2953}_{\pm0.006}$ | $\mathbf{0.8689}_{\pm0.005}$ | $\mathbf{0.2658}_{\pm0.007}$ | $\mathbf{0.8268}_{\pm0.006}$ | $\mathbf{0.2258}_{\pm0.006}$ | $\mathbf{0.7567}_{\pm0.004}$ | $\mathbf{0.1936}_{\pm0.005}$ |
| | NP | $0.7643_{\pm0.015}$ | $0.2041_{\pm0.015}$ | $0.7342_{\pm0.002}$ | $0.1725_{\pm0.008}$ | $0.6892_{\pm0.004}$ | $0.1542_{\pm0.006}$ | $0.6235_{\pm0.008}$ | $0.1342_{\pm0.007}$ |
| | BNP | $0.8156_{\pm0.005}$ | $0.2689_{\pm0.007}$ | $0.7789_{\pm0.004}$ | $0.2215_{\pm0.005}$ | $0.7421_{\pm0.005}$ | $0.2117_{\pm0.007}$ | $0.6715_{\pm0.006}$ | $0.1828_{\pm0.006}$ |
| | SNP | $\mathbf{0.8368}_{\pm0.006}$ | $\mathbf{0.2844}_{\pm0.005}$ | $\mathbf{0.8036}_{\pm0.004}$ | $\mathbf{0.2483}_{\pm0.006}$ | $\mathbf{0.7635}_{\pm0.004}$ | $\mathbf{0.2325}_{\pm0.006}$ | $\mathbf{0.6973}_{\pm0.006}$ | $\mathbf{0.2016}_{\pm0.005}$ |
| | ANP | $1.2421_{\pm0.002}$ | $0.6366_{\pm0.004}$ | $1.2115_{\pm0.001}$ | $0.6001_{\pm0.008}$ | $1.1784_{\pm0.004}$ | $0.5622_{\pm0.006}$ | $1.1252_{\pm0.007}$ | $0.5274_{\pm0.008}$ |
| | BANP | $1.3456_{\pm0.003}$ | $0.6514_{\pm0.005}$ | $1.3125_{\pm0.005}$ | $0.6115_{\pm0.002}$ | $1.2672_{\pm0.004}$ | $0.5711_{\pm0.005}$ | $1.2236_{\pm0.006}$ | $0.5306_{\pm0.006}$ |
| | SANP | $\mathbf{1.3721}_{\pm0.002}$ | $\mathbf{0.6653}_{\pm0.004}$ | $\mathbf{1.3461}_{\pm0.003}$ | $\mathbf{0.6256}_{\pm0.004}$ | $\mathbf{1.3011}_{\pm0.003}$ | $\mathbf{0.5782}_{\pm0.004}$ | $\mathbf{1.2457}_{\pm0.005}$ | $\mathbf{0.5356}_{\pm0.002}$ |

## C.2  System Identification on Physics Engines

The second synthetic experiment focuses on evaluating model dynamics on a classical simulator, Cart-Pole systems, which is detailed in [7, 27]. The Cart-Pole swing-up task is a standard benchmark for nonlinear control due to the non-linearity in the dynamics, and the requirement for nonlinear controllers to successfully swing up and balance the pendulum. A pendulum of length $l$ is attached to a cart by a frictionless pivot. The system begins with the cart at position $x_c = 0$ and the pendulum hanging down: $\theta$. The goal is to accelerate the cart by applying horizontal force $u_t$ at each time-step $t$ to invert and then stabilize the pendulum's endpoint at the goal. There are some parameters that need to be known, such as cart mass $m_c$, pendulum mass $m_p$, acceleration of gravity $g = 9.82 m/s^2$, time horizon $T$, time discretization $\triangle t$ and ground friction coefficient $f_c$. In this case, the Cart-Pole swing-up task aims to forecast the transited state $[x_c, \theta, x'_c, \theta']$ in time step $t + 1$ based on the input as a state action pair $[x_c, \theta, x'_c, \theta', a]$ in time step $t$.

For system identification task on physics engines, the neural architectures for CNP, NP, ANP, BANP and our RNP refer to Appendix B. The number of hidden unites is $d_h = 32$ and latent representation $d_z = 32$. The number of layers are $l_e = l_{de} = l_{la} = l_{qk} = l_v = 2$.

To generate a variety of trajectories under a random policy for this experiment, the mass mc and the ground friction coefficient $f_c$ are varied in the discrete choices $m_c \in \{0.3, 0.4, 0.5, 0.6, 0.7\}$ and $f_c \in \{0.06, 0.08, 0.1, 0.12\}$. Each pair of $[m_c, f_c]$ values specifies a dynamics environment, and we formulate all pairs of $m_c \in \{0.3, 0.5, 0.7\}$ and $f_c \in \{0.08, 0.12\}$ as training environments with the rest 16 pairs of configurations as the testing environments. For each configuration of the simulator including training and testing environments, we sample 400 trajectories of horizon as 10 steps using a random controller, and more details refer to Supplementary material. During the testing process,

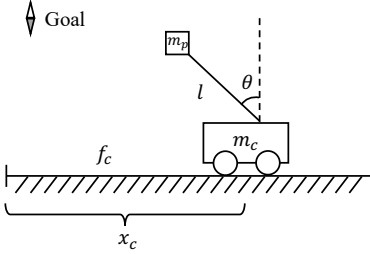

Figure 4: Cart-Pole Dynamical Systems. The cart and the pole are with masses $m_c$ and $m_p$, and the length of the pole is $l$. And the configuration of the simulator is up to parameters of the cart-pole mass and the ground friction coefficient here with other hyper-parameters fixed in this experiment.

Table 9: Average Log-likelihoods over all context and target points on EMNIST and CELEBA.

| Dataset | Method | Original | | Noise(+5%) | | Noise(+10%) | | Noise(+15%) | |
|---|---|---|---|---|---|---|---|---|---|
| | | context | target | context | target | context | target | context | target |
| EMNIST | CNP | $0.9522_{\pm0.023}$ | $0.7515_{\pm0.0015}$ | $0.8977_{\pm0.0016}$ | $0.6336_{\pm0.017}$ | $0.8242_{\pm0.0018}$ | $0.5784_{\pm0.009}$ | $0.6566_{\pm0.0017}$ | $0.5341_{\pm0.016}$ |
| | BCNP | $0.9678_{\pm0.010}$ | $0.8058_{\pm0.008}$ | $0.9015_{\pm0.008}$ | $0.6711_{\pm0.009}$ | $0.8415_{\pm0.007}$ | $0.6089_{\pm0.009}$ | $0.6788_{\pm0.006}$ | $0.5715_{\pm0.006}$ |
| | SCNP | $\mathbf{0.9716}_{\pm0.008}$ | $\mathbf{0.8343}_{\pm0.006}$ | $\mathbf{0.9251}_{\pm0.008}$ | $\mathbf{0.6971}_{\pm0.007}$ | $\mathbf{0.8674}_{\pm0.006}$ | $\mathbf{0.6343}_{\pm0.007}$ | $\mathbf{0.6986}_{\pm0.005}$ | $\mathbf{0.5877}_{\pm0.005}$ |
| | NP | $0.9678_{\pm0.004}$ | $0.7756_{\pm0.005}$ | $0.9011_{\pm0.009}$ | $0.6941_{\pm0.006}$ | $0.8544_{\pm0.009}$ | $0.6455_{\pm0.007}$ | $0.7034_{\pm0.009}$ | $0.5865_{\pm0.006}$ |
| | BNP | $0.9757_{\pm0.005}$ | $0.8358_{\pm0.005}$ | $0.8116_{\pm0.007}$ | $0.7625_{\pm0.006}$ | $0.8759_{\pm0.007}$ | $0.6773_{\pm0.007}$ | $0.7451_{\pm0.005}$ | $0.6237_{\pm0.005}$ |
| | SNP | $\mathbf{0.9847}_{\pm0.005}$ | $\mathbf{0.8562}_{\pm0.006}$ | $\mathbf{0.8368}_{\pm0.006}$ | $\mathbf{0.7844}_{\pm0.005}$ | $\mathbf{0.8984}_{\pm0.005}$ | $\mathbf{0.6984}_{\pm0.005}$ | $\mathbf{0.7653}_{\pm0.005}$ | $\mathbf{0.6456}_{\pm0.004}$ |
| | ANP | $1.1125_{\pm0.002}$ | $1.0321_{\pm0.004}$ | $0.9815_{\pm0.002}$ | $0.6366_{\pm0.006}$ | $0.9021_{\pm0.004}$ | $0.7053_{\pm0.008}$ | $0.8454_{\pm0.002}$ | $0.7034_{\pm0.005}$ |
| | BANP | $1.1355_{\pm0.003}$ | $1.0615_{\pm0.005}$ | $1.0236_{\pm0.002}$ | $0.6549_{\pm0.005}$ | $0.9155_{\pm0.004}$ | $0.7521_{\pm0.006}$ | $0.8612_{\pm0.003}$ | $0.7515_{\pm0.005}$ |
| | SANP | $\mathbf{1.1531}_{\pm0.000}$ | $\mathbf{1.0877}_{\pm0.004}$ | $\mathbf{1.0421}_{\pm0.002}$ | $\mathbf{0.6776}_{\pm0.005}$ | $\mathbf{0.9321}_{\pm0.002}$ | $\mathbf{0.7843}_{\pm0.006}$ | $\mathbf{0.8732}_{\pm0.003}$ | $\mathbf{0.7657}_{\pm0.005}$ |
| CELEBA | CNP | $1.0323_{\pm0.016}$ | $0.7845_{\pm0.013}$ | $1.0177_{\pm0.016}$ | $0.7438_{\pm0.017}$ | $0.8956_{\pm0.009}$ | $0.7344_{\pm0.011}$ | $0.7677_{\pm0.012}$ | $0.6096_{\pm0.009}$ |
| | BCNP | $1.0452_{\pm0.009}$ | $0.8015_{\pm0.008}$ | $1.0275_{\pm0.009}$ | $0.7726_{\pm0.008}$ | $0.9351_{\pm0.006}$ | $0.8376_{\pm0.009}$ | $0.8015_{\pm0.010}$ | $0.6816_{\pm0.008}$ |
| | SCNP | $\mathbf{1.0525}_{\pm0.008}$ | $\mathbf{0.8243}_{\pm0.006}$ | $\mathbf{1.0348}_{\pm0.008}$ | $\mathbf{0.7868}_{\pm0.006}$ | $\mathbf{0.9562}_{\pm0.004}$ | $\mathbf{0.8545}_{\pm0.008}$ | $\mathbf{0.8344}_{\pm0.006}$ | $\mathbf{0.7045}_{\pm0.005}$ |
| | NP | $1.1333_{\pm0.004}$ | $0.8766_{\pm0.005}$ | $1.1043_{\pm0.015}$ | $0.8355_{\pm0.015}$ | $1.0034_{\pm0.008}$ | $0.8456_{\pm0.006}$ | $0.8935_{\pm0.009}$ | $0.6893_{\pm0.006}$ |
| | BNP | $1.1732_{\pm0.005}$ | $0.8901_{\pm0.006}$ | $1.1378_{\pm0.007}$ | $0.8678_{\pm0.006}$ | $1.0411_{\pm0.008}$ | $0.8711_{\pm0.005}$ | $0.9256_{\pm0.008}$ | $0.7671_{\pm0.006}$ |
| | SNP | $\mathbf{1.1952}_{\pm0.005}$ | $\mathbf{0.9062}_{\pm0.006}$ | $\mathbf{1.1565}_{\pm0.005}$ | $\mathbf{0.8846}_{\pm0.005}$ | $\mathbf{1.0542}_{\pm0.006}$ | $\mathbf{0.8956}_{\pm0.004}$ | $\mathbf{0.9425}_{\pm0.006}$ | $\mathbf{0.7985}_{\pm0.005}$ |
| | ANP | $1.2633_{\pm0.002}$ | $1.0163_{\pm0.004}$ | $1.1377_{\pm0.004}$ | $0.9866_{\pm0.006}$ | $1.0418_{\pm0.004}$ | $0.8845_{\pm0.006}$ | $0.9363_{\pm0.004}$ | $0.7346_{\pm0.008}$ |
| | BANP | $1.2751_{\pm0.002}$ | $1.0389_{\pm0.005}$ | $1.1488_{\pm0.004}$ | $1.0155_{\pm0.005}$ | $1.0602_{\pm0.005}$ | $0.9255_{\pm0.006}$ | $0.9489_{\pm0.004}$ | $0.8415_{\pm0.007}$ |
| | SANP | $\mathbf{1.2854}_{\pm0.000}$ | $\mathbf{1.0594}_{\pm0.004}$ | $\mathbf{1.1685}_{\pm0.002}$ | $\mathbf{1.0353}_{\pm0.004}$ | $\mathbf{1.0772}_{\pm0.002}$ | $\mathbf{0.9455}_{\pm0.004}$ | $\mathbf{0.9655}_{\pm0.003}$ | $\mathbf{0.8673}_{\pm0.005}$ |

100 state transition pairs are randomly selected for each configuration of the environment, working as the maximum context points to identify the configuration of dynamics.

### C.3 Image Completion

Analogous to the 1D experiments, we take random pixels of a given image at training as targets, and select a subset of this as contexts, again choosing the number of contexts and targets randomly ($n \sim U[3, 200]$, $m \sim n + U[0, 200 - n]$). The $\mathbf{x}_i$ are rescaled to $[-1, 1]$ and the $y_i$ are rescaled to $[-0.5, 0.5]$. We use a batch size of 16 for both EMNIST and CelebA, i.e. use 16 randomly selected images for each batch.

For image completion experiments on EMNIST and CelebA dataset, the neural architectures for CNP, NP, ANP, BCNP, BNP, BANP, and our SCNP, SNP, and SANP refer to Appendix B. The number of hidden unites is $d_h = 128$ and latent representation $d_z = 128$. The number of layers are $l_e = l_{de} = 4, l_{la} = l_{qk} = l_v = 5$. $h_{head} = 8$

### C.4 Uncertainty Measuring

Methods for reasoning under uncertainty are a key building block of accurate and reliable machine learning systems. We further analyze the learned models using the framework introduced in [17] to quantify uncertainty by investigate the calibration error and sharpness of the models. By assuming predictive distribution $P_\theta(y_i|\mathbf{x}_i, \mathcal{D}^{\mathcal{C}})$ as Gaussian distribution $\mathcal{N}(y_i; \mu_i, \sigma_i^2)$, we can get the probabilistic forecast $F_i(x_i)$. More formally, $m$ confidence levels $0 \leq p_1 < p_2 < \cdots p_m \leq 1$ are choose. For each threshold $p_j$, we compute the empirical frequency

$$\hat{p}_j = \frac{|y_i|F_i(y_i) \leq p_j, i \in \mathcal{T}|}{|\mathcal{T}|} \tag{31}$$

Table 10: Calibration error and sharpness of the models for 1D regression experiments (Mean $\pm$ Std).

| Method | RBF | | Matern | |
|---|---|---|---|---|
| | CAL | SHA | CAL | SHA |
| CNP | $0.0724_{\pm0.008}$ | $0.0887_{\pm0.007}$ | $0.0514_{\pm0.005}$ | $0.0831_{\pm0.010}$ |
| BCNP | $0.1015_{\pm0.007}$ | $0.1151_{\pm0.005}$ | $0.0724_{\pm0.005}$ | $0.1152_{\pm0.009}$ |
| SCNP | $0.1225_{\pm0.005}$ | $0.1357_{\pm0.006}$ | $0.0425_{\pm0.005}$ | $0.1325_{\pm0.008}$ |
| NP | $0.0615_{\pm0.004}$ | $0.0715_{\pm0.005}$ | $0.0343_{\pm0.015}$ | $0.0717_{\pm0.015}$ |
| BNP | $0.0871_{\pm0.005}$ | $0.1052_{\pm0.006}$ | $0.0325_{\pm0.009}$ | $0.0715_{\pm0.008}$ |
| SNP | $0.1155_{\pm0.004}$ | $0.1257_{\pm0.005}$ | $0.0347_{\pm0.008}$ | $0.0718_{\pm0.007}$ |
| ANP | $0.1532_{\pm0.002}$ | $0.0616_{\pm0.004}$ | $0.0921_{\pm0.004}$ | $0.0871_{\pm0.006}$ |
| BANP | $0.2353_{\pm0.002}$ | $0.0689_{\pm0.004}$ | $0.0752_{\pm0.005}$ | $0.0741_{\pm0.006}$ |
| SANP | $0.2633_{\pm0.000}$ | $0.0741_{\pm0.004}$ | $0.0415_{\pm0.002}$ | $0.0667_{\pm0.004}$ |

Table 11: Calibration error and sharpness of the models for system identification experiments (Mean $\pm$ Std).

| Method | CAL | SHA |
|---|---|---|
| CNP | $0.0872_{\pm0.008}$ | $0.0415_{\pm0.007}$ |
| BCNP | $0.1051_{\pm0.005}$ | $0.0765_{\pm0.006}$ |
| SCNP | $0.1124_{\pm0.005}$ | $0.0833_{\pm0.005}$ |
| NP | $0.0821_{\pm0.003}$ | $0.0581_{\pm0.005}$ |
| BNP | $0.0952_{\pm0.003}$ | $0.0616_{\pm0.005}$ |
| SNP | $0.1001_{\pm0.001}$ | $0.0668_{\pm0.005}$ |
| ANP | $0.0863_{\pm0.001}$ | $0.0673_{\pm0.004}$ |
| BANP | $0.1235_{\pm0.001}$ | $0.0815_{\pm0.005}$ |
| SANP | $0.1431_{\pm0.000}$ | $0.1094_{\pm0.004}$ |

In this case, the calibration error is defined as a numerical score describing the quality of forecast calibration:

$$\text{CAL}(F_1, y_1, \cdots, F_{|\mathcal{T}|}, y_{|\mathcal{T}|}) = \sum_{j=1}^{m} w_j \cdot (p_j - \hat{p}_j)^2 \tag{32}$$

here $w_j$ is weight and we set $w_j = 1$.

Sharpness is measured by using the variance $var(F_i) = \sigma_i^2$ of the random variable whose CDF is $F_i$. Low-variance predictions are tightly centered around one value. A sharpness score can be defined by

$$\text{SHA}(F_1, \cdots, F_{|\mathcal{T}|}) = \frac{1}{|\mathcal{T}|} \sum_{i \in \mathcal{T}} \sigma_i^2 \tag{33}$$

We evaluated the CE and sharpness of CNP, NP, ANP, BCNP, NBP, BANP, and corresponding stable versions SCNP, SNP, and SANP trained in the experiments. Table 10, 11, 12 list the calibration error and sharpness score on 1D regression, system identification, and image completion tasks. In several settings, models with our stable solution can achieve better calibration and sharpness but work worse in some settings, such as the calibration error being worse than NP and ANP in terms of 1D regression tasks with Matern. The possible reason is that our method tends to produce conservative credible intervals, so become under-confident or less over-confident in some settings.

Table 12: Calibration error and sharpness of the models for image completion experiments (Mean $\pm$ Std).

| Method | EMNIST | | CELEBA | |
|---|---|---|---|---|
| | CAL | SHA | CAL | SHA |
| CNP | $0.0182_{\pm0.002}$ | $0.0574_{\pm0.002}$ | $0.0253_{\pm0.002}$ | $0.0743_{\pm0.001}$ |
| BCNP | $0.0415_{\pm0.003}$ | $0.0716_{\pm0.002}$ | $0.0412_{\pm0.002}$ | $0.0981_{\pm0.002}$ |
| SCNP | $0.0543_{\pm0.001}$ | $0.0846_{\pm0.002}$ | $0.0457_{\pm0.002}$ | $0.1136_{\pm0.002}$ |
| NP | $0.0163_{\pm0.001}$ | $0.0671_{\pm0.002}$ | $0.0261_{\pm0.001}$ | $0.0711_{\pm0.001}$ |
| BNP | $0.0352_{\pm0.002}$ | $0.0815_{\pm0.002}$ | $0.0463_{\pm0.001}$ | $0.0981_{\pm0.002}$ |
| SNP | $0.0446_{\pm0.002}$ | $0.0918_{\pm0.000}$ | $0.0532_{\pm0.001}$ | $0.1046_{\pm0.001}$ |
| ANP | $0.0156_{\pm0.002}$ | $0.0656_{\pm0.002}$ | $0.0261_{\pm0.004}$ | $0.0815_{\pm0.006}$ |
| BANP | $0.0412_{\pm0.002}$ | $0.0871_{\pm0.002}$ | $0.0513_{\pm0.003}$ | $0.1125_{\pm0.003}$ |
| SANP | $0.0558_{\pm0.000}$ | $0.0954_{\pm0.001}$ | $0.0632_{\pm0.002}$ | $0.1265_{\pm0.004}$ |

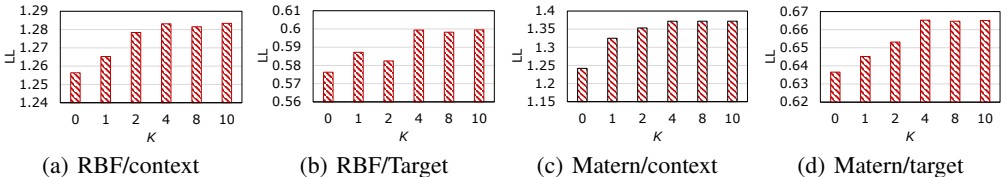

Figure 5: The Log-likelihood comparisons with different $K$ on 1D regression task.

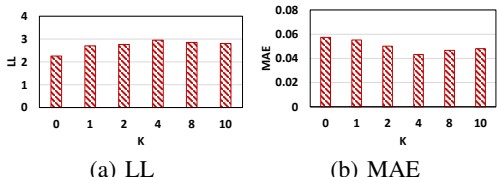

Figure 6: The LL and MAE comparisons with different $K$ on system identification task.

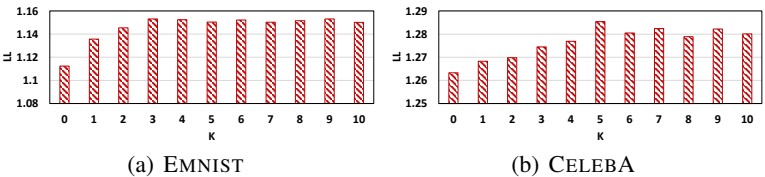

Figure 7: The LL comparisons with different $K$ on image completion task.

## C.5 Ablation Study

The key parameter in our stable solution is the number of hard predictive subsets $K$. Taking SANP as an example, we investigated the average log-likelihood in terms of different $K$ on 1D regression task, as shown in Figure 5. We can see that SANP performs better as $K$ increases, reaches the best value at around $K = 4$, and then becomes stable in performance as $K$ grows larger. As proved in Theorem 5.4 in the main manuscript, optimization on $\mathcal{D}^{\mathcal{C}}$ and more than one hard predicted subsets of $\mathcal{D}^{\mathcal{C}}$ can achieve more stable prediction. Similarly, we also conducted experiments to investigate the effect of $K$ on System identification and image completion task and similar observations can be seen in Figure 6 and 7.

