# OpenReview forum: "Neural Processes with Stability"
_NeurIPS.cc/2023/Conference — NeurIPS 2023 poster_

### Official Review · Reviewer_Gzsw · 2023-07-01

**Soundness:** 3 good
**Presentation:** 4 excellent
**Contribution:** 3 good
**Rating:** 7
**Confidence:** 4

**Summary:**

This paper proposes a new way of training neural processes that increases their algorithmic stability, a property known to be helpful for generalisation in a frequentist sense. They work out theory to motivate their procedure, which is then compared against other existing ways of training neural processes. They show their method improves performance on a variety of tasks.

**Strengths:**

- The paper is well-written and well-motivated
- The method seems to show consistent improvements in a wide variety of tasks


**Weaknesses:**

- I think the authors could spend more time arguing for the importance of stability. I agree that it would be helpful, but it's not clear to me that it is the single crucial factor in making neural processes generalise.
- The authors try of the NP and ANP, but could consider more SOTA methods such as the convolutional neural process.

**Questions:**

- The method shows consistent improvement in quantitative metrics, but when looking at Fig 2 it is difficult to see why this is the case. It's not clear to me why the SNP predictions are such a significant improvement over the NP. Could the authors give a qualitative explanation why? If not, I'm concerned that the improvement in numbers could be an artefact of something not genuinely significant.

**Limitations:**

The authors give a very small amount of the paper to address limitations. I would expect a more extensive discussion here. In particular, how does the runtime of their algorithm compare with regular NPs? Is there some kind of trade-off here?

---

> ### Author Rebuttal · Authors · 2023-08-09
>
> Thanks for your valuable feedback. We made efforts to address your every concern and question. If we have any misunderstanding or further questions, please feel free to let us know and we will reply quickly.
>
> **Q1 The importance of stability**
>
> > Thank you for your suggestion. This suggestion is very important for us to improve the motivation of the paper. Based on the empirical risk and expected risk of neural processes defined in Section 3 (note that due to the unique meta-learning mechanism of NP, this type of empirical risk and expected risk is different from that of general machine learning models), we define the algorithmic stability of neural processes in detail in Section 4 and explain what kind of model is more stable. Based on this, we give the experiments of NP on stability and generalization error and use examples to prove the relationship between model generalization error and stability. We did not say that stability is the only key factor for NP to generalize, we just described their relationship, and thus gave the theoretical analysis based on stability.
> >
> > For your question, we will also add the following expressions about the importance of stability in the problem introduction part. The sparsity of the data, incomplete and noisy introduces challenges to the algorithm stability. NP models are biased to the quality of context data and target data, so small changes in the training data (noisy) may significantly change the models. In this case, unstable solutions will introduce high training error variance, and minimizing the training error may not guarantee consistent error reduction on the testing dataset, i.e., low generalization performance. In other words, the algorithm stability has a direct impact on generalization performance, and an unstable NP solution has low generalization performance.
>
> **Q2 Could consider more SOTA methods such as the convolutional neural process**
>
> > Thank you for your suggestions. We conducted experiments on ConvCNP and ConvNP, and built our stable versions, SConvCNP and SConvNP, based on them. The table below lists the results on RBF kernel on 1D regression (due to word limit, we only list the results on original data and Noise(+15%)). By comparing the results in the table below with those in the main text, we can see that ConvCNP and ConvNP results are better than other traditional NPs (including CNP, NP, and ANP). In addition, the stable versions SConvCNP and SConvNP results are better than the original ConvCNP and ConvNP, which also confirms the effectiveness of our weighting strategy on ConvCNP and ConvNP.
> >
> > We will add the results of ConvCNP, ConvNP, SConvCNP, and SConvNP on various tasks in the new version.
> >
> > | RBF Kernel | Original-Context | Original-Target | Noise(+15%)-Context | Noise(+15%)-Target |
> > | :--------: | :--------------: | :-------------: | :-----------------: | :----------------: |
> > |  ConvCNP   |      1.2631      |     0.6421      |       1.0241        |       0.4262       |
> > |   ConvNP   |      1.2874      |     0.6503      |       1.0523        |       0.4311       |
> > |  SConvCNP  |      1.3991      |     0.6793      |       1.1641        |       0.4396       |
> > |  SConvNP   |      1.4036      |     0.6831      |       1.1825        |       0.4425       |
>
>
> **Q3 About the question of Fig 2**
>
> > We apologize for the confusion caused by the content shown in Fig 2. In fact, Fig 2 (a) and (c) only show the visualization of the mean and variance of 50 samples randomly selected from one task in NP and SNP under different kernels (RBF and Matern). It can be seen that in this example, both NP and SNP fit the context data well, while for the target data, some parts of SNP have better fitting characteristics than NP. One thing worth noting is that SNP and NP have similar variances when fitting the target without context information. SNP may produce slightly larger variances to indicate the uncertainty of this part of information. For some deterministic regions, SNP will produce smaller variances to indicate the certainty of this part of information. Although it may be difficult to see the significant advantages of SNP from a single visualization result, in fact, the overall fitting result of SNP is better than NP.
>
> **Q4 Limitations**
>
> > We pointed out the limitations of the proposed theoretical proof in the submission version. That is, it only applies to regression tasks, and its validity for classification tasks remains to be verified. In fact, as you mentioned, computational complexity is also a point worth paying attention to. As shown in Algorithm 1, we need to pre-train the base model to select samples. In fact, the pre-trained model can not only be used for sample selection, but its model parameters can be used as the initialization of the stable version model. At this time, the training of the stable version can converge faster. Taking the 1D regression task as an example, the traditional NP model needs to train 200 epochs using 256 batches per epoch of batch size 16 to get the final result. When using our stable strategy, in order to train and obtain the pre-trained model, only 100 epochs are needed to select the easily predicted subset, and relying on the pre-trained model parameters as the initialization of the stable version, only 100 epochs are also needed to converge to a satisfactory result. Therefore, it can be said that the computational complexity of NPs with our stable solution is similar to the original NPs.
> >
> > Although the number of training epochs of our model is similar to that of existing methods, it incurs $K$ times additional computational cost during training compared to other models. Thus, in addition to the applicability problem of the theoretical proof that has been pointed out, we will also have a more detailed discussion on the limitation of computational complexity.
>
>
> Finally, thanks again for your valuable comments.

---

> > ### Comment · Reviewer_Gzsw · 2023-08-12
> >
> > Thanks for your response and especially for new experiments on the ConvCNP and ConvNP. I would like to further clarify regarding the computational cost of stable NPs: you mention that it incurs $K$ times additional cost during training. Does it incur any additional cost during evaluation time compared to "non-stable" NPs?

---

> > > ### Author Response · Authors · 2023-08-14
> > >
> > > Thanks for your feedback. Our model incurs $K$ times additional computational cost during training compared to other models, as it requires calculating a new predictive distribution with $K-1$ augmented dataset (See Eq. (10). Therefore, it can be seen that the only difference is in the loss function compared to the original model, so the computational cost is only increased when calculating the loss, and the model parameters are essentially the same as the original model. Usually, we can use parallel strategies to calculate the loss on $K-1$ sub-datasets, thereby reducing the impact of this part. In addition, there is one more point we would like to emphasize, that is, as shown in Algorithm 1, we need to pre-train the base model to select samples. In fact, the pre-trained model can not only be used for sample selection, but its model parameters can be used as the initialization of the stable version model. At this time, the training of the stable version can converge faster. Taking the 1D regression task as an example, the traditional NP model needs to train 200 epochs using 256 batches per epoch of batch size 16 to get the final result. When using our stable strategy, in order to train and obtain the pre-trained model, only 100 epochs are needed to select the easily predicted subset, and relying on the pre-trained model parameters as the initialization of the stable version, only 100 epochs are also needed to converge to a satisfactory result. Therefore, it can be said that the computational complexity of NPs with our stable solution is similar to the original NPs. I hope my reply answered your question, thank you again for your feedback.

---

> > > > ### Comment · Reviewer_Gzsw · 2023-08-17
> > > >
> > > > Thanks for the clarification. I presume this means that at inference time the cost of the forward pass for stable NPs and regular NPs is identical?
> > > >
> > > > I am maintaining my score as Accept and increasing confidence to 4.

---

> > > > > ### Author Response · Authors · 2023-08-18
> > > > >
> > > > > Thank you very much for your recognition. Yes, the stable version and the original version have similar inference times. The following table shows the comparison of training and testing time between CNP and SCNP under different tasks. In terms of training time, SCNP’s time includes the traditional NP training time for the first 100 epochs, as well as the stable strategy training time for 100 epochs based on traditional NP parameter initialization. It can be seen that SCNP requires more additional training time than traditional CNP because stable strategy requires additional $K$ augmented terms , but the overall increase in time is acceptable. In fact, the computation of these $K$ items can be improved by some parallel strategies in essence. It is gratifying that in terms of testing time, it can be seen that the original version of NP and the stable version of NP have similar testing times, which means that the stable version of NP also inherits the ability of neural process to quickly infer the shape of a new function at test time.
> > > > >
> > > > > |      | RBF-train [Seconds (in thousands)] | RBF-test [Seconds] | Cart-Pole-train [Seconds (in thousands)] | Cart-Pole-test [Seconds] | EMNIST-train [Seconds (in thousands)] | EMNIST-test [Seconds] |
> > > > > | ---- | :--------------------------------: | :----------------: | :--------------------------------------: | :----------------------: | :-----------------------------------: | :-------------------: |
> > > > > | CNP  |        23.451$_{\pm0.003}$         | 5.243$_{\pm0.23}$  |           13.527$_{\pm0.002}$            |    3.535$_{\pm0.16}$     |         341.531$_{\pm0.001}$          |  16.321$_{\pm0.15}$   |
> > > > > | SCNP |        28.522$_{\pm0.003}$         | 5.251$_{\pm0.20}$  |           15.523$_{\pm0.003}$            |    3.512$_{\pm0.15}$     |         369.815$_{\pm0.001}$          |  16.274$_{\pm0.13}$   |
> > > > >
> > > > > Thank you again for your recognition and further discussion.

---

### Official Review · Reviewer_i5nU · 2023-07-04

**Soundness:** 2 fair
**Presentation:** 3 good
**Contribution:** 3 good
**Rating:** 5
**Confidence:** 5

**Summary:**

This paper focuses on the scenario where noisy data is included in the context points during the training of Neural Processes. In order to address algorithmic instability, the authors provide theoretical guidelines to obtain stable solutions that exhibit high generalization capabilities. These guidelines use hard predictable subsets as additional different training datasets.

**Strengths:**

Originality

- This paper introduces numerous theoretical findings regarding stable solutions.
- The idea is overall interesting and novel.

Clarity
- The paper is effectively written and presents its content in a clear and comprehensible manner.

**Weaknesses:**

Experiment
- The choice of baseline models appears to be somewhat outdated. Considering the existence of newer NP models after Attentive Neural Processes [1] and Bootstrapping Neural Processes [2], it would be beneficial to include more recent approaches such as Convolutional Neural Processes [3,4] as baselines.
- The tasks employed in the experiments seem to be too simplistic. The results obtained from simple 1D regression and 2D regression datasets may not be sufficient to demonstrate the superiority of the proposed model. I recommend conducting tasks such as Bayesian Optimization and the Predator-Prey task from [2] to showcase the general applicability of the method. Additionally, it would be valuable to include results using a periodic kernel to address model-data mismatch scenarios, as the Matern kernel's similarity to the RBF kernel may not adequately demonstrate the robustness of the proposed model.


References

[1] H. Kim, A. Mnih, J. Schwarz, M. Garnelo, S. M. A. Eslami, D. Rosenbaum, and V. Oriol. Attentive neural processes. In International Conference on Learning Representations (ICLR), 2018.

[2] J. Lee, Y. Lee, J. Kim, E. Yang, S. J. Hwang, and Y. W. Teh. Bootstrapping neural processes. In Advances in Neural Information Processing Systems 33 (NeurIPS 2020), 2020.

[3] J. Gordon, W. P. Bruinsma, A. Y. K. Foong, J. Requeima, Y. Dubois, and R. E. Turner. Convolutional conditional neural processes. In International Conference on Learning Representations (ICLR), 2020.

[4] A. Y. K. Foong, W. P. Bruinsma, J. Gordon, Y. Dubois, J. Requeima, and R. E. Turner. Metalearning stationary stochastic process prediction with convolutional neural processes. In Advances in Neural Information Processing Systems 33 (NeurIPS 2020), 2020.

**Questions:**

Experiment
- In Algorithm 1, it appears that the NP model requires early training to classify hard and easy datasets. I'm curious to know the number of early training epochs needed for the NP model. Additionally, it would be helpful to understand if the training epochs for other baselines are the same as those for the proposed method. This would help ascertain whether the performance gain stems from additional training or the effectiveness of the proposed method itself.
- Could you provide separate log likelihood results for the context dataset, target dataset, and the entire dataset (referred to as "target" in this paper) by splitting them? In reference [1] and [2], log likelihoods of the context and target are reported separately, and increasing the context log likelihood can sometimes lead to an increase in the whole log likelihood without any actual performance improvement (or even a decrease in performance) for the true target dataset.
- It would be valuable to conduct experiments using other baselines and additional tasks as mentioned earlier. The current set of experiments, including 1D regression with RBF kernel, 2D regression, and the physics engine experiment, may not be sufficient to demonstrate the superiority of the proposed model across various tasks.
- How are the values of 'w' in the loss function balanced? Could you perform an ablation study on these hyperparameters to determine the robustness of the training procedure with respect to these hyperparameters?

References

[1] H. Kim, A. Mnih, J. Schwarz, M. Garnelo, S. M. A. Eslami, D. Rosenbaum, and V. Oriol. Attentive neural processes. In International Conference on Learning Representations (ICLR), 2018.

[2] J. Lee, Y. Lee, J. Kim, E. Yang, S. J. Hwang, and Y. W. Teh. Bootstrapping neural processes. In Advances in Neural Information Processing Systems 33 (NeurIPS 2020), 2020.

**Limitations:**

Limitation
- The proposed model incurs K times additional computational cost during training compared to other models, as it requires calculating a new predictive distribution with K-1 augmented dataset.

---

> ### Author Rebuttal · Authors · 2023-08-09
>
> Thanks for your valuable feedback. We made efforts to address your every concern and question. If we have any misunderstanding or further questions, please feel free to let us know and we will reply quickly.
>
> **Q1 Additional baselines including ConvCNP and ConvNP**
>
> > Thank you for your suggestions. We conducted experiments on ConvCNP and ConvNP, and built our stable versions, SConvCNP and SConvNP, based on them. The table below lists the results on RBF for 1D regression (due to word limit, we only list the results on original data and Noise(+15%)). By comparing the results in the table below with those in the main text, we can see that ConvCNP and ConvNP results are better than other traditional NPs (including CNP, NP, and ANP). In addition, the stable versions SConvCNP and SConvNP results are better than the original ConvCNP and ConvNP, which also confirms the effectiveness of our weighting strategy on ConvCNP and ConvNP.
> >
> > We will add the results of ConvCNP, ConvNP, SConvCNP, and SConvNP on various tasks in the new version.
> >
> >
> > | RBF Kernel | Original-Context | Original-Target | Noise(+15%)-Context | Noise(+15%)-Target |
> > | :--------: | :--------------: | :-------------: | :-----------------: | :----------------: |
> > |  ConvCNP   |      1.2631      |     0.6421      |       1.0241        |       0.4262       |
> > |   ConvNP   |      1.2874      |     0.6503      |       1.0523        |       0.4311       |
> > |  SConvCNP  |      1.3991      |     0.6793      |       1.1641        |       0.4396       |
> > |  SConvNP   |      1.4036      |     0.6831      |       1.1825        |       0.4425       |
>
> **Q2 Additional tasks including  Bayesian Optimization and the Predator-Prey task**
>
> > Following the setting in BANP (Bootstraping attentive neural process), we conducted the Bayesian optimization experiment. Taking GP data with RBF and Matern priors as examples, we gave the results of ANP, BANP, and our SANP. See Figure 1 in the uploaded pdf file for details. It can be seen that SANP has better overall performance.
> >
> > Following the setting in ConvCNP and BANP, we conducted experiments on the Predator-prey task. We first trained the models using simulated data generated from a Lotka-Volterra model and tested on real-world data (Hudson’s Bay hare-lynx data).  Due to the response limitation, we list the results in Table 1 of the uploaded pdf file.  Similar to the previous observation, our stable version still outperforms the original version. Among stable versions, SConvNP achieves the best performance.
>
> **Q3 Additional kernel (periodic) in the 1D regression task**
>
> > Thank you for your suggestions. We conducted experiments on 1D regression tasks with Periodic kernel. Following the setting of BANP and similar noise settings of our previous kernels, we list the results on both original data and noise(+15) data in Table 2 of the uploaded pdf file.  In this model-data mismatch data, stable versions still significantly outperform their corresponding original versions.
> >
> > We will add those results in the new version.
>
> **Q4 The number of training epochs**
>
> > Thank you for your good suggestions. As shown in Algorithm 1, we need to pre-train the base model to select samples. In fact, the pre-trained model can not only be used for sample selection, but its model parameters can be used as the initialization of the stable version model. At this time, the training of the stable version can converge faster. Taking the 1D regression task as an example, the traditional NP model needs to train 200 epochs using 256 batches per epoch of batch size 16 to get the final result. When using our stable strategy, in order to train and obtain the pre-trained model, only 100 epochs are needed to select the easily predicted subset, and relying on the pre-trained model parameters as the initialization of the stable version, only 100 epochs are also needed to converge to a satisfactory result.
> >
> > We have made some explanations in Section 5.1 of the original version, and we will make a more detailed description to explain the computational loss of the stable model compared to the traditional model.
>
> **Q5 Separate log likelihood results for the context dataset, target dataset, and the entire dataset**
>
> > Thank you for your suggestion. This point you mentioned is very important, so here we assume that the target set $\mathcal{T} = \{1,2,\cdots, N\} - \mathcal{C}$, and calculate the log-likelihood on the context dataset, target dataset, and the entire dataset. Table 3 of the uploaded pdf file lists the results for 1D regression with RBF kernel.  It can be seen that the log-likelihood of the true target is significantly improved under the action of the stable strategy. In fact, in other experimental settings, the stable version has a significant improvement over the base version on the target dataset. This also proves that the proposed stable solution can improve the predictive performance on the target dataset. We will distinguish the log-likelihood under different target settings in the new version.
>
>  **Q6 The values of weights $w$**.
>
> > In the experimental setting, we only set $w_k = \frac{1}{K+1} \forall k=0,1,2,\cdots,K$ to satisfy the requirement of $w_0+w_1+\cdots+w_K=1$ in the theorem. In fact, we performed an ablation study on $K$ in the main text and the appendix to investigate how our stable solution performs by varying the number of  subsets $K$.  It should be noted that, when $K = 0$, the stable model is degraded to the original base NP model. Thus, the fact that the stable solution can achieve better performance than the original model with additional terms $\sum_{k=1}^K w_k R_{D_{1:M}^k}(\theta)$ . In addition, the log-likelihood in all cases increases as $K$ increases. This further confirms Theorem 5.4: probing easily predictable entries to form harder predictable entry sets can better increase the model performance.
>
> Finally, thanks again for your valuable comments.

---

> > ### Comment · Reviewer_i5nU · 2023-08-15
> >
> > I appreciate your thorough explanations and the additional experiments you've provided. However, I still have a lingering question regarding Question 6. Theorem 5.4 states that any set of combinations for $w$ values, as long as they fulfill the condition $w_1+\cdots+w_K=1$, will result in an enhancement of performance. Through this ablation study, I aim to empirically validate the practical validity of this theorem. Furthermore, I'm interested in observing the impact of altering the $w$ values on the final outcomes while keeping the value of K constant.

---

> > > ### Author Response · Authors · 2023-08-18
> > >
> > > Thank you very much for your experimental suggestions and further discussion on the weight $w_k$. Actually, we assume that $w_0+w_1+\cdots+w_K=1$ since we can scale it with a scaler if $w_0+w_1+\cdots+w_K \neq 1$. . When we set $w_k = \frac{1}{K+1}$, it is essentially set that each module contributes equally. We conducted different experiments to explore the impact of different $w_k$ . Taking the 1D regression task with RBF-GP data as an example, we set $K=3$ and different $w_k$ for experiments. The data in the first row of the table below is the set weight, and the information in parentheses indicates the corresponding proportion of other weights when $w_0=1$. Here, target likelihood is the true target likelihood in question 3. It can be seen from the table below that when different weights are set, the model using stable strategy is better than the original model, and when the weight is set to be equal, its performance is optimal. In addition, when the weight of $w_i (i\geq1)$ is significantly different from $w_0$, such as (0.625, 0.125, 0.125, 0.125) and (0.0625, 0.3125, 0.3125, 0.3125), its performance is more significantly reduced compared to (0.25, 0.25 , 0.25, 0.25), but it still has a significant improvement compared to the original non-stable model.
> > >
> > > | Model    | $0.25,0.25,0.25,0.25$  $(1,1,1,1)$ | $0.4,0.2,0.2,0.2$ $(1,0.5,0.5,0.5)$ | $0.625,0.125,0.125,0.125$ $(1,0.2,0.2,0.2)$ | $0.1,0.3,0.3,0.3$ $(1,3,3,3)$ | $0.0625, 0.3125,0.3125,0.3125$ $(1,5,5,5)$ |
> > > | -------- | :--------------------------------: | :---------------------------------: | :-----------------------------------------: | :---------------------------: | :----------------------------------------: |
> > > | CNP      |          (0.8724, 0.3953)          |          (0.8724, 0.3953)           |              (0.8724, 0.3953)               |       (0.8724, 0.3953)        |              (0.8724, 0.3953)              |
> > > | SCNP     |        **(0.9255, 0.4125)**        |          (0.9127, 0.4019)           |              (0.9035, 0.3998)               |       (0.9149, 0.4086)        |              (0.8927, 0.3991)              |
> > > | NP       |          (0.8215, 0.3426)          |          (0.8215, 0.3426)           |              (0.8215, 0.3426)               |       (0.8215, 0.3426)        |              (0.8215, 0.3426)              |
> > > | SNP      |        **(0.8955, 0.3925)**        |          (0.8737, 0.3817)           |              (0.8716, 0.3809)               |       (0.8831, 0.3859)        |              (0.8657, 0.3775)              |
> > > | ANP      |          (1.2563, 0.5052)          |          (1.2563, 0.5052)           |              (1.2563, 0.5052)               |       (1.2563, 0.5052)        |              (1.2563, 0.5052)              |
> > > | SANP     |        **(1.2831, 0.5215)**        |          (1.2776, 0.5187)           |              (1.2738, 0.5196)               |       (1.2791, 0.5203)        |              (1.2712, 0.5193)              |
> > > | ConvCNP  |          (1.2631, 0.5871)          |          (1.2631, 0.5871)           |              (1.2631, 0.5871)               |       (1.2631, 0.5871)        |              (1.2631, 0.5871)              |
> > > | SConvCNP |        **(1.3991, 0.5996)**        |          (1.3916, 0.5933)           |              (1.3841, 0.5915)               |       (1.3934, 0.5946)        |              (1.3854, 0.5919)              |
> > > | ConvNP   |          (1.2874, 0.5904)          |          (1.2874, 0.5904)           |              (1.2874, 0.5904)               |       (1.2874, 0.5904)        |              (1.2874, 0.5904)              |
> > > | SConvNP  |        **(1.4036, 0.6015)**        |          (1.3991, 0.6004)           |              (1.3931, 0.5992)               |       (1.4012, 0.6015)        |              (1.3957, 0.5995)              |
> > >
> > > I hope my reply can answer your questions. Thank you again for your advice and further discussion.

---

> > > > ### Comment · Reviewer_i5nU · 2023-08-19
> > > >
> > > > I will raise my evaluation score accordingly. Please incorporate all the extra experiments and corresponding discussions into the updated manuscript.

---

> > > > > ### Author Response · Authors · 2023-08-21
> > > > >
> > > > > Thank you very much for your feedback. We will carefully consider your suggestions and incorporate all the extra experiments and discussions into the updated manuscript.

---

### Official Review · Reviewer_jySn · 2023-07-05

**Soundness:** 1 poor
**Presentation:** 3 good
**Contribution:** 1 poor
**Rating:** 3
**Confidence:** 4

**Summary:**

In this paper, the stability of neural processes was studied. The scenario is when the context data points are noisy and this paper introduces the concept of stability, proposes adaptive strategies and draw some conclusions from empirical observations.

**Strengths:**

This paper is written in a neat way and the consideration of noisy context points can be necessary in some risk sensitive applications. The literature work is well overviewed.

**Weaknesses:**

Even though this work is easy to follow, there exists unfixable issues concerning the research motivation, definitions of stability and derived theorems.

To begin with, let us rethink types of uncertainty in neural processes. *As we know, there primarily exist two types of uncertainty, namely aleatoric uncertainty and epistemic uncertainty.* The considered noisy context points belong to the aleatoric type, which is irreducible. The number of context points or observation extent corresponds to the epistemic, which means the prediction is more accurate with increasing the number of context points.

**1. Concept of algorithmic stability in Definition4.1.** In Line 141, the concept of stability is presented, and it suggests that the approximation results remain stable if the change of the datasets given a stable NP (see Line178-179). These definition and conclusion violate the property of predictive functions in stochastic processes, where increase of context points decreases the epistemic uncertainty.
Particularly, **in Line133, the stable property defined in NPs is “A stable learning algorithm has the property  that replacing one element in the training set does not result in a significant change to the algorithm’s output”.** **I strongly disagree with this trait since NPs are Bayesian  non-parametric, which means the change of context points definitely changes the predictive distribution like that in Gaussian processes. This property enables the Bayesian optimization using NPs.** If the proposed stability is achieved, the crucial Bayesian non-parametric property no longer holds.


**2. Connection between subset selection and noisy context data points.** It is unclear in a theoretical sense that how the subset selection alleviate the negative effect of noisy context points. This work considers the scenario of noise context points but focuses on the algorithmic stability solved by the bootstrapping trick. Hence, the logics are not well organized.

**3. Overstated or imprecise claims.** In line 29-30, to motivate this work, it says “As demonstrated in previous work [7, 8, 12, 9], existing NPs cannot provide stable predictions under noisy conditions. However, as far as I know, not all of these work mentioned the importance of noisy conditions and did investigations in an empirical and theoretical way. In Line 101, it states that the ELBO is maximized in NPs, however, the real optimization objective is the approximate ELBO rather than real ELBO since the Jessen inequality no longer holds.


**Questions:**

No other questions.

**Limitations:**

See the weakness section.

---

> ### Author Rebuttal · Authors · 2023-08-09
>
> Thanks for your valuable feedback. We made efforts to address your every concern and question. If we have any misunderstanding or further questions, please feel free to let us know and we will reply quickly.
>
>
>
> **Q1 Concept of algorithmic stability in Definition 4.1.**
>
> > Thank you for your suggestions. In fact, the definition of algorithm stability we gave in Definition 4.1 is based on the difference between the empirical risk and the expected risk of NP, rather than the accuracy of a certain type of prediction. Therefore, we hope that when the model is disturbed, especially by noise disturbance, it can still be close to the expected risk. And what we said when an element in the dataset changes, especially when it is disturbed or irrelevantly changed, the approximate result of NP is still stable, which does not contradict what you said about uncertainty. The example of Gaussian processes you gave “the change of context points definitely changes the predictive distribution like that in Gaussian processes” refers to the change of prediction results, while we define the change of NP model from the perspective of stability (the difference between expected risk and empirical risk), that is, when changing a sample or an element in the dataset, the model result will not have a large disturbance.
>
>
>
> **Q2 Connection between subset selection and noisy context data points.**
>
> > This paper discusses algorithmic stability, and algorithmic stability is strongly related to the noise data. For this reason, we gave the algorithm stability definition applicable to NP in the problem description in Section 4. Based on this, we explored the relationship between stability and generalization error by relying on the experiment of replacing noise data and thus introduced the use of stability definition to prove the impact of subset selection on NP stability. The experimental results also proved that the subset selection strategy has significant improvements compared to the baselines under different noise conditions.
>
>
>
> **Q3 Overstated or imprecise claims.**
>
> > What we are expounding here is that the results of existing work can show that classical models are easily affected by noise, especially under fewer context samples, the results of different models are very different, and they are far from the correct results. For this reason, we listed a few classic references to illustrate, such as CNP, NP, ANP, and ConvCNP. Here, we will make a more detailed description and reference adjustment. In addition, you mentioned the ELBO of NP in line 101. Usually, we can express the objective function in the form of ELBO (such as Eq. 3 in the original ANP paper), which is also a common expression for the objective function of VAE type.
>
>
>
> Finally, thanks again for your valuable comments.

---

> > ### Comment · Reviewer_jySn · 2023-08-16
> >
> > I would like to appreciate the author's response. However, the definition of algorithmic stability contradicts my cognition about epistemic uncertainty. As this also relates to the subset selection, I'm still unconvinced by the reply. As the result, I keep the score.

---

> > > ### Author Response · Authors · 2023-08-18
> > >
> > > We are very sorry that our reply did not satisfy you. Thank you again for discussing the paper.

---

### Official Review · Reviewer_nxGu · 2023-07-09

**Soundness:** 3 good
**Presentation:** 3 good
**Contribution:** 2 fair
**Rating:** 4
**Confidence:** 3

**Summary:**

This work studies the algorithmic stability and generalization error of neural processes (NPs). Motivated by the stability analysis of NPs, a new method is proposed to train NPs that results in improved stability and better generalization performance with theoretical guarantees. The proposed method is based on the idea of removing easy datasets from the meta-dataset, which is essentially a weighted-training algorithm that down weights easy samples. The authors also empirically verify the usefulness of the proposed method on both synthetic and real-world datasets.

**Strengths:**

1. The idea of studying the algorithmic stability and generalization error of NPs is interesting and well-motivated.
2. The proposed method for improving the robustness of NPs has theoretical guarantees and appears to be useful in practice.
3. The paper is well-written and easy to follow.

**Weaknesses:**

1. In the beginning of the proof for every theorem, the random variables $R_{\tau}(\theta)-(...)$ are assume to have zero mean with no justifications. It is unclear why they have zero means. The authors should justify these assumptions since all proofs are based on them.
2. The proposed method is essentially a weighted-training algorithm that down weights easy datasets. According to the definition in the paper, a dataset is easy if the NLL of *every data point* in it is smaller than $R_{\mathcal{D}_{1:M}}(\theta)$ under a regularly trained NP. I can imagine that in practice each dataset will have some easy data points and some hard data points, but it is less likely that all data points in a data set are easy. The authors should discuss the applicability of the proposed method in this regard.
3. The proposed method is computationally more expensive, since it requires 1) training a regular NP; 2) filtering out easy datasets according to some criteria; 3) reweighting each dataset; and 4) training the final NP. The extra computational costs will especially be enlarged when the NP is used for downstream tasks such as Bayesian optimization where one needs to constantly update the model as new data points/datasets arrive. The authors should discuss and compare the computational costs between the proposed method and regular NPs.
4. It would be more convincing if the authors can show that the NPs trained by the proposed method can lead to improved performance in downstream tasks such as Bayesian optimization.

**Questions:**

See the weaknesses section. I will increase my score accordingly if some or all of the weakness points are addressed.

**Limitations:**

Limitations are not discussed in the paper.

This work does not have obvious potential negative societal impact.

---

> ### Author Rebuttal · Authors · 2023-08-09
>
> Thanks for your valuable feedback. We made efforts to address your every concern and question. If we have any misunderstanding or further questions, please feel free to let us know and we will reply quickly.
>
>
>
> **Q1 The definition of the random variable with zero mean in theorems.**
>
> > The random variables we assume are the difference between the expected risk and the empirical risk, and considering the accuracy of the model, we expect the empirical risk to be as close as possible to the empirical risk, hence the zero-mean assumption.
> >
> > We will explain the setting of this assumption in more detail in subsequent versions.
>
>
>
> **Q2 The definition of easy datasets.**
>
> > We are not defining a dataset as a simple dataset, as shown in section 5.1, we are selecting samples that are easy to predict from M meta-datasets to form an easily predicted subset, thus we are not defining a certain dataset as an easy dataset.
>
>
>
> **Q3 Discuss and compare the computational costs between the proposed method and regular NPs.**
>
> > Thank you for your good suggestions. As shown in Algorithm 1, we need to pre-train the base model to select samples. In fact, the pre-trained model can not only be used for sample selection, but its model parameters can be used as the initialization of the stable version model. At this time, the training of the stable version can converge faster. Taking the 1D regression task as an example, the traditional NP model needs to train 200 epochs using 256 batches per epoch of batch size 16 to get the final result. When using our stable strategy, in order to train and obtain the pre-trained model, only 100 epochs are needed to select the easily predicted subset, and relying on the pre-trained model parameters as the initialization of the stable version, only 100 epochs are also needed to converge to a satisfactory result. Therefore, it can be said that the computational complexity of NPs with our stable solution is similar to the original NPs.
> >
> > We have made some explanations in Section 5.1 of the original version, and we will make a more detailed description to explain the computational loss of the stable model compared to the traditional model.
> >
> >
>
> **Q4 Performance comparison in downstream tasks such as Bayesian optimization.**
>
> > Following the setting in BANP (Bootstraping attentive neural process), we conducted the Bayesian optimization experiment. Taking GP data with RBF and Matern priors as examples, we gave the results of ANP, BANP, and our SANP. Please see Figure 1 in the uploaded pdf file for details. It can be seen that SANP has better overall performance.
>
> Finally, thanks again for your valuable comments.

---

> > ### Comment · Reviewer_nxGu · 2023-08-16
> >
> > Thanks for your response!
> >
> > Your clarifications have addressed my first two concerns. Apologies for the misunderstandings.
> >
> > Regarding the computational costs, it would be more convincing if you could provide quantitative comparisons (e.g., wall time) for the original NP and your stable solution averaged over multiple runs with different random seeds on at least two experiments conducted in the paper. Currently, it is unclear whether the result you described for the 1D regression task is consistent across multiple runs with different random seeds and/or whether other experiments will have similar results.
> >
> > Thanks for conducting the additional Bayesian optimization experiments. However, the improvement of SANP over other baselines seems to be inconsistent: the results on data generated by RBF-GP look good, but SANP underperforms BANP quite a lot on data generated by Matern-GP.

---

> > > ### Author Response · Authors · 2023-08-18
> > >
> > > Thank you very much for your advice and further discussion.
> > >
> > > **Q1 Wall time comparison**
> > >
> > > > The following table shows the comparison of training and testing time between CNP and SCNP under different tasks. In terms of training time, SCNP’s time includes the traditional NP training time for the first 100 epochs, as well as the stable strategy training time for 100 epochs based on traditional NP parameter initialization. It can be seen that SCNP requires more additional training time than traditional CNP because stable strategy requires additional $K$ augmented terms , but the overall increase in time is acceptable. In fact, the computation of these $K$ items can be improved by some parallel strategies in essence. It is gratifying that in terms of testing time, it can be seen that the original version of NP and the stable version of NP have similar testing time, which means that the stable version of NP also inherits the ability of neural process to quickly infer the shape of a new function at test time.
> > > >
> > > > | Model | RBF-train [Seconds (in thousands)] | RBF-test [Seconds] | Cart-Pole-train [Seconds (in thousands)] | Cart-Pole-test [Seconds] | EMNIST-train [Seconds (in thousands)] | EMNIST-test [Seconds] |
> > > > | ----- | :--------------------------------: | :----------------: | :--------------------------------------: | :----------------------: | :-----------------------------------: | :-------------------: |
> > > > | CNP   |        23.451$_{\pm0.003}$         | 5.243$_{\pm0.23}$  |           13.527$_{\pm0.002}$            |    3.535$_{\pm0.16}$     |         341.531$_{\pm0.001}$          |  16.321$_{\pm0.15}$   |
> > > > | SCNP  |        28.522$_{\pm0.003}$         | 5.251$_{\pm0.20}$  |           15.523$_{\pm0.003}$            |    3.512$_{\pm0.15}$     |         369.815$_{\pm0.001}$          |  16.274$_{\pm0.13}$   |
> > >
> > > **Q2 Bayesian optimization experiments on data generated by Matern-GP**
> > >
> > > > For Matern-GP, we listed a randomly selected data for visualization in the uploaded pdf. This set of results shows that SANP still has performance improvement compared with ANP, but is slightly worse than BANP. We found that this may be caused by the set of results we randomly selected. We found that although BANP can have better results on some data, its overall results are more differentiated, that is, its stability deviation. Although our SANP is slightly worse than BANP in the visualized example, its overall performance is still better than SANP and traditional ANP. The following table lists the numerical results on three types of data (we also conducted experiments on data generated on Periodic-GP), and it can be seen that SANP is better than ANP and BANP.
> > > >
> > > > |      | RBF-GP              | Matern-GP           | Periodic-GP         |
> > > > | ---- | ------------------- | ------------------- | ------------------- |
> > > > | ANP  | 1.1245$_{\pm0.003}$ | 1.1518$_{\pm0.003}$ | 1.1892$_{\pm0.002}$ |
> > > > | BANP | 0.1341$_{\pm0.003}$ | 1.1316$_{\pm0.004}$ | 1.1788$_{\pm0.005}$ |
> > > > | SANP | 0.1142$_{\pm0.002}$ | 1.1201$_{\pm0.002}$ | 1.1672$_{\pm0.001}$ |
> > >
> > >
> > >
> > > I hope my reply can answer your questions. Thank you again for your advice and further discussion.

---

### Author Rebuttal · Authors · 2023-08-09

Dear Reviewers, AC and SAC,

We would like to thank the reviewers for their valuable feedback and insightful comments and AC and SAC for their efforts in the review work. In this rebuttal, we address the main concerns and answered the questions raised by each reviewer individually.

---

### Decision · Program_Chairs · 2023-09-21

**Decision:**

Accept (poster)

**Comment:**

This paper introduces an approach to enhance the predictive performance and robustness of neural process families by incorporating algorithmic stability considerations. The method is designed to be seamlessly integrated with various types of neural processes, demonstrating significant improvements in accuracy and robustness across a variety of out-of-distribution scenarios.

While the average review score falls below the threshold for acceptance, I would like to highlight that the current score might not accurately represent the overall sentiment of the reviews. Notably, reviewer nXGu (borderline reject) initially raised concerns regarding formulation errors and inference time slowdown, which the authors appear to have effectively addressed during the rebuttal phase. However, these improvements do not seem to have been fully reflected in the final review score. Reviewer jYSn (reject) expressed reservations about the stability concept, leading to their recommendation for rejection. However, I align more closely with the viewpoint of Reviewer Gzsw (advocating acceptance) on this particular issue.

In summary, I find the proposed method to be versatile, applicable across a wide spectrum of neural process models. The negative assessments from the current reviews do not appear to definitively warrant rejection of the paper. As such, my recommendation is to accept it.